# ADAPTIVE RETRIEVAL AND SCALABLE INDEXING FOR $k$-NN SEARCH WITH CROSS-ENCODERS

**Nishant Yadav**[1,*]**, Nicholas Monath**[2]**, Manzil Zaheer**[2]**, Rob Fergus**[2]**, Andrew McCallum**[1]

[1] University of Massachusetts Amherst, [2] Google DeepMind

## ABSTRACT

Cross-encoder (CE) models which compute similarity by jointly encoding a query-item pair perform better than using dot-product with embedding-based models (dual-encoders) at estimating query-item relevance. Existing approaches perform $k$-NN search with cross-encoders by approximating the CE similarity with a vector embedding space fit either with dual-encoders (DE) or CUR matrix factorization. DE-based retrieve-and-rerank approaches suffer from poor recall as DE generalizes poorly to new domains and the test-time retrieval with DE is decoupled from the CE. While CUR-based approaches can be more accurate than the DE-based retrieve-and-rerank approach, such approaches require a prohibitively large number of CE calls to compute item embeddings, thus making it impractical for deployment at scale. In this paper, we address these shortcomings with our proposed sparse-matrix factorization based method that efficiently computes latent query and item representations to approximate CE scores and performs $k$-NN search with the approximate CE similarity. In an offline indexing stage, we compute item embeddings by factorizing a sparse matrix containing query-item CE scores for a set of train queries. Our method produces a high-quality approximation while requiring only a fraction of CE similarity calls as compared to CUR-based methods, and allows for leveraging DE models to initialize the embedding space while avoiding compute- and resource-intensive finetuning of DE via distillation. At test time, we keep item embeddings fixed and perform retrieval over multiple rounds, alternating between a) estimating the test query embedding by minimizing error in approximating CE scores of items retrieved thus far, and b) using the updated test query embedding for retrieving more items in the next round. Our proposed $k$-NN search method can achieve up to 5% and 54% improvement in $k$-NN recall for $k = 1$ and 100 respectively over the widely-used DE-based retrieve-and-rerank approach. Furthermore, our proposed approach to index the items by aligning item embeddings with the CE achieves up to $100\times$ and $5\times$ speedup over CUR-based and dual-encoder distillation based approaches respectively while matching or improving $k$-NN search recall over baselines.

## 1 INTRODUCTION

Efficient and accurate nearest neighbor search is paramount for retrieval (Menon et al., 2022; Rosa et al., 2022; Qu et al., 2021), classification in large output spaces (e.g., entity linking (Ayoola et al., 2022; Logeswaran et al., 2019; Wu et al., 2020)), non-parametric models (Das et al., 2022; Wang et al., 2022), and many other such applications in machine learning (Goyal et al., 2022; Izacard et al., 2023; Bahri et al., 2020). The accuracy and efficiency of nearest neighbor search depends on a combination of factors (1) the computational cost of pairwise distance comparisons between datapoints, (2) preprocessing time for constructing a nearest neighbor index (e.g., dimensionality reduction (Indyk, 2000), quantization (Ge et al., 2013; Guo et al., 2020), data structure construction (Beygelzimer et al., 2006; Malkov & Yashunin, 2018; Zaheer et al., 2019)), and (3) the time taken to query the index to retrieve the nearest neighbor(s).

Similarity functions such as cross-encoders which take a pair of data points as inputs and directly output a scalar score, have achieved state-of-the-art results on numerous tasks (e.g., QA (Qu et al.,

---
*Now at Google DeepMind

2021; Thakur et al., 2021b), entity linking (Logeswaran et al., 2019)). However, these models are exceptionally computationally expensive since these are typically parameterized by several layers of neural models such as transformers (Vaswani et al., 2017), and scoring each item for a given query requires a forward pass of the large parametric model, making them impractical similarity functions to use directly in nearest neighbor indices (Yadav et al., 2022). Initial work has approximated search with cross-encoders (CE) for a given test query using a heuristic retrieve-and-rerank approach that uses a separate model to retrieve a subset of items followed by re-ranking using the CE. Prior work performs the initial retrieval using dot-product of sparse query/item embedding from models such as BM25, or dense query/item embeddings from models such as dual-encoders (DE) which are typically trained on the same task and data as the CE. To support search with CE, recent work (Yadav et al., 2022; 2023) improves upon heuristic retrieve-and-rerank approaches, by directly learning an embedding space that approximates the CE score function. These approaches use CUR decomposition (Mahoney & Drineas, 2009) to compute (relatively) low-dimensional embeddings for queries and items. The item embeddings are computed by scoring each item against a set of *anchor/train* queries. At test-time, the test query embedding is computed by using CE scores of the test query against a set of (adaptively-chosen) *anchor* items.

Both DE-based retrieve-and-rerank and CUR-based methods are not well suited for a typical application setting in $k$-NN search – building an index on a new set of targets with a given (trained) similarity function. The DE-based approach has several disadvantages in this setting. DE models show poor generalization to new domains and thus require additional fine-tuning on the target domain to improve performance Yadav et al. (2022); Thakur et al. (2021a). This can be both resource-intensive as well time-consuming. Furthermore, it requires access to the parameters (not just embedding outputs) of the DE, which might not be possible if the DE is provided by an API service. On the other hand, while CUR-based approaches outperform retrieve-and-rerank approaches without additional fine-tuning of DE, they require computing a dense score matrix by scoring each item against a set of anchor/train queries. This does not scale well with the number of items. For instance, for a domain with 500 anchor/train queries and 10K items, it takes around 10 hours[1] to compute the dense query-item score matrix with a CE parameterized using `bert-base` (Yadav et al., 2022). By simple extrapolation, indexing 5 million items using 500 queries would take around 5000 GPU hours.

In this paper, we propose a sparse-matrix factorization-based approach to improve the efficiency of fitting an embedding space to approximate the cross-encoder for $k$-NN search. Our proposed approach significantly reduces the offline indexing cost as compared to existing approaches by constructing a sparse matrix containing cross-encoder scores between a set of training queries ($\mathcal{Q}_{\text{train}}$) and all the items ($\mathcal{I}$), and using efficient matrix factorization methods to produce a set of item embeddings that are aligned with the cross-encoder. At test-time, our proposed approach, AXN, computes a test query embedding to approximate cross-encoder scores between the test query and items, and performs retrieval using approximate cross-encoder scores. AXN performs retrieval over multiple rounds while keeping the item embedding fixed and incrementally refining the test query embedding using cross-encoder scores of the items retrieved in previous rounds. In the first round, the cross-encoder is used to score the test query against a small number of items chosen uniformly at random or baseline retrieval methods such as dual-encoder or BM25. In each subsequent round, AXN alternates between (a) updating the test query embedding to improve the approximation of the cross-encoder score of items retrieved so far, and (b) retrieving additional items using the improved approximation of the cross-encoder, and computing the exact cross-encoder scores for the retrieved items. Finally, the retrieved items are ranked based on exact cross-encoder scores and the top-$k$ items returned as the $k$-nearest neighbors for the given test query.

We perform an empirical evaluation of our method using cross-encoder models trained for the task of entity linking and information retrieval on ZESHEL (Logeswaran et al., 2019) and BEIR (Thakur et al., 2021b) benchmark respectively. Our proposed $k$-NN search method can be used together with dense item embeddings produced by any method such as baseline dual-encoder models and still yield up to 5% and 54% improvement in $k$-NN recall for $k$ =1 and 100 respectively over retrieve-and-rerank style inference with the same dual-encoder. Furthermore, our proposed approach to align item embeddings with the cross-encoder achieves up to $100\times$ and $5\times$ speedup over CUR-based approaches and training dual-encoders via distillation-based respectively while matching or improving test-time $k$-NN search recall over baseline approaches.

---

[1]On an Nvidia 2080ti GPU with 12 GB memory using batch size=50

## 2 PROPOSED APPROACH

**Task Description** A cross-encoder model $f : \mathcal{Q} \times \mathcal{I} \to \mathbb{R}$ maps a query-item pair $(q, i) \in \mathcal{Q} \times \mathcal{I}$ to a scalar similarity. We consider the task of similarity search with the cross-encoder, in particular finding the $k$-nearest neighbors items for a given query $q$ from a fixed set of items $\mathcal{I}$:

$$\mathcal{N}(q) \triangleq \underset{i \in \mathcal{I}}{\arg \operatorname{top} k} \, f(q, i) \tag{1}$$

where $\arg \operatorname{top} k$ returns the indices of the top $k$ scoring items of the function. Exact $k$-NN search with a cross-encoder would require $\mathcal{O}(|\mathcal{I}|)$ cross-encoder calls as an item needs to be jointly encoded with the test query in order to compute its score. Since cross-encoders are typically parameterized using deep neural models such as transformers (Vaswani et al., 2017), $\mathcal{O}(|\mathcal{I}|)$ calls to the cross-encoder model can be prohibitively expensive at test time. Therefore, we tackle the task of approximate $k$-NN search with cross-encoder models. Let $\hat{f}(\cdot, \cdot)$ denote the approximation to the cross-encoder that is learned using exact cross-encoder scores for a sample of query-item pairs. We refer to the approximate $k$-nearest neighbors as $\hat{\mathcal{N}}(q) \triangleq \arg \operatorname{top} k_{i \in \mathcal{I}} \, \hat{f}(q, i)$ and measure the quality of the approximation using nearest neighbor recall: $\frac{|\hat{\mathcal{N}}(q) \cap \mathcal{N}(q)|}{|\mathcal{N}(q)|}$

In this work, we assume black-box access to the cross-encoder[2], access to the set of items and train queries from the target domain, and a base dual-encoder ($\text{DE}_{\text{SRC}}$) trained on the same task and source data as the cross-encoder. In §2.1, we first present our proposed sparse-matrix factorization based method to compute item embeddings in an offline step. In §2.2, we present an online approach to compute a test query embedding to approximate the cross-encoder scores and perform $k$-NN search using the approximate cross-encoder scores.

### 2.1 PROPOSED OFFLINE INDEXING OF ITEMS

In this section, we describe our proposed approach to efficiently align the item embeddings with the cross-encoder where efficiency is measured in terms of the number of training samples (query-item pairs) required to be gathered and scored using the cross-encoder and wall-clock time to fit an approximation of the cross-encoder model. We consider an approximation of the cross-encoder with an inner-product space where a query ($q$) and an item ($i$) are represented with $d$-dimensional vectors $\mathbf{u}_q \in \mathbb{R}^d$ and $\mathbf{v}_i \in \mathbb{R}^d$ respectively. $k$-NN search using this approximation corresponds to solving the following vector-based nearest neighbor search:

$$\hat{\mathcal{N}}(q) \triangleq \underset{i \in \mathcal{I}}{\arg \operatorname{top} k} \, \mathbf{u}_q \mathbf{v}_i^\mathsf{T}. \tag{2}$$

This vector-based $k$-nearest neighbor search can potentially be made more efficient using data structures such as cover trees (Beygelzimer et al., 2006), HNSW (Malkov & Yashunin, 2018), or any of the many other highly effective vector nearest neighbor search indexes (Guo et al., 2020; Johnson et al., 2019). The focus of our work is not on a new way to make the vector nearest neighbor search more efficient, but rather to develop efficient and accurate methods of fitting the embedded representations of $\mathbf{u}_q$ and $\mathbf{v}_i^\mathsf{T}$ to approximate the cross-encoder scores.

Let $G \in \mathbb{R}^{|\mathcal{Q}_{\text{train}}| \times |\mathcal{I}|}$ denote the pairwise similarity matrix containing the exact cross-encoder over the pairs of training queries ($\mathcal{Q}_{\text{train}}$) and items ($\mathcal{I}$). We assume that $G$ is *partially observed* or incomplete, that is only a very small subset of the query-item pairs ($\mathcal{P}_{\text{train}}$) are observed in $G$. Let $U \in \mathbb{R}^{|\mathcal{Q}_{\text{train}}| \times d}$ and $V \in \mathbb{R}^{|\mathcal{I}| \times d}$ be matrices such that each row corresponds to the embedding of a query $q \in \mathcal{Q}_{\text{train}}$ and an item $i \in \mathcal{I}$ respectively. We optimize the following widely-used objective for matrix completion to estimate $U$ and $V$ via stochastic gradient descent:

$$\min_{U \in \mathbb{R}^{|\mathcal{Q}_{\text{train}}| \times d}, V \in \mathbb{R}^{|\mathcal{I}| \times d}} \|(G - UV^\mathsf{T})_{\mathcal{P}_{\text{train}}}\|_2 \tag{3}$$

where $(\cdot)_{\mathcal{P}_{\text{train}}}$ denotes projection on the set of observed entries in $G$. There are two important considerations: (1) how to select with values of $G$ to observe (and incur the cost of running the cross-encoder model), and (2) how to compute/parameterize the matrices $U$ and $V$.

---

[2]Approximating a neural scoring function by compressing, approximating, quantizing the scoring function is widely studied but outside the scope of this paper.

**Constructing Sparse Matrix** $G$    Given a set of items ($\mathcal{I}$) and train queries ($\mathcal{Q}_{\text{train}}$), we construct the sparse matrix $G$ by selecting $k_d$ items $\mathcal{I}_q \subset \mathcal{I}$ for each query $q \in \mathcal{Q}_{\text{train}}$ either uniformly at random or using top-$k_d$ items from a baseline retrieval method such as the base dual-encoder ($\text{DE}_{\text{SRC}}$). This approach requires $k_d|\mathcal{Q}_{\text{train}}|$ calls to the cross-encoder. We also experiment with an approach that selects $k_d$ queries $\mathcal{Q}_i \subset \mathcal{Q}_{\text{train}}$ for each item $i \in \mathcal{I}$, and thus requires $k_d|\mathcal{I}|$ calls to the cross-encoder.

**Parameterizing and Training** $U$ **and** $V$

- **Transductive** ($\text{MF}_{\text{TRNS}}$): In this setting, $U$ and $V$ are trainable parameters and are learned by optimizing the objective in Eq. 4. $U$ and $V$ can be optionally initialized using query and item embeddings from the base dual-encoder ($\text{DE}_{\text{SRC}}$). Note that this parameterization requires scoring each item against at least a small number of queries to update the embedding of an item from its initialized value, thus requiring scoring of $\mathcal{O}(|\mathcal{I}|)$ query-item pairs to construct the sparse matrix $G$. Such an approach may not scale well with the number of items as the number of cross-encoder calls to construct $G$ and the number of trainable parameters are both linear in the number of items. For instance, when $|\mathcal{I}|$ = 5 million, $d = 1000$, $V$ would contain 5 billion trainable parameters.

- **Inductive** ($\text{MF}_{\text{IND}}$): In this setting, we train parametric models to produce query and item embeddings $U$ and $V$ from (raw) query and item features such as textual descriptions of queries and items. Unlike transductive approaches, inductive matrix factorization approaches can produce embeddings for unseen queries and items, and thus can be used to produce embeddings for items not scored against any train query in matrix $G$ as well as embeddings for test queries $q_{\text{test}} \notin \mathcal{Q}_{\text{train}}$. Prior work typically uses $\text{DE}_{\text{SRC}}$ (a DE trained on the same task and source domains as the CE) and finetunes $\text{DE}_{\text{SRC}}$ on the target domain via distillation using the CE. However, training all parameters of such parametric encoding models via distillation can be compute- and resource-intensive as these models are built using several layers of neural models such as transformers. Recall that our goal is to efficiently build an accurate approximation of the CE on a given target domain. Thus, to improve the efficiency of fitting the approximation of the CE, we propose to train a shallow MLP model (using data from the target domain) that takes query/item embeddings from $\text{DE}_{\text{SRC}}$ as input and outputs updated embeddings while keeping $\text{DE}_{\text{SRC}}$ parameters frozen.

## 2.2 Proposed Test-Time $k$-NN Search Method: AxN

At test-time, we need to perform $k$-NN search for a test query $q_{\text{test}} \notin \mathcal{Q}_{\text{train}}$, and thus need to compute an embedding for the test query in order to approximate cross-encoder scores and perform retrieval with the approximate scores. Note that computing the test query embedding by factorizing the matrix $G$ at *test-time* while including the test query $q_{\text{test}}$ is computationally infeasible. Thus, an ideal solution would be to compute item representations in an offline indexing step, and compute the test query embedding *on-the-fly* while keeping item embeddings fixed. A potential solution is to use a parametric model such as $\text{DE}_{\text{SRC}}$ or $\text{MF}_{\text{IND}}$ to compute test query embedding, perform retrieval using inner-product scores between test query and item embeddings, and finally, re-rank the retrieved items using the cross-encoder. While such a retrieve-and-rerank approach can work, the retrieval step on such an approach is decoupled from the re-ranking model, and thus may result in poor recall.

In this work, we propose an adaptive approach AxN, which stands for "**A**daptive **C**ross-Encoder **N**earest Neighbor Search". As described in Algorithm 1, AxN performs retrieval over $\mathcal{R}$ rounds while incrementally refining the cross-encoder approximation for $q_{\text{test}}$ by updating $\mathbf{u}_{q_{\text{test}}}$, the embedding for $q_{\text{test}}$. The test-time inference latency (and throughput) depends largely on the number of cross-encoder calls made at test time as each cross-encoder call requires a forward pass through a large neural model. Thus, we operate under a fixed computational budget which allows for up to $\mathcal{B}_{\text{CE}}$ cross-encoder calls at test-time.

Let $\mathcal{A}_r$ be the cumulative set of items chosen up to round $r$. In the first round ($r = 1$), we select $\mathcal{B}_{\text{CE}}/\mathcal{R}$ items either uniformly at random or using separate retrieval models such as dual-encoders or BM25 and compute the exact cross-encoder scores of these items for the given test query. We compute the test query embedding $\mathbf{u}_{q_{\text{test}}}$ by solving the following system of linear equations

$$V_{\mathcal{A}_r}\mathbf{u}_{q_{\text{test}}} = \mathbf{a}_r \tag{4}$$

where $V_{\mathcal{A}_r} \in \mathbb{R}^{|\mathcal{A}_r| \times d}$ contains embeddings for items in $\mathcal{A}_r$, and $\mathbf{a}_r$ contains cross-encoder scores for $q_{\text{test}}$ paired with items in $\mathcal{A}_r$. In round $r > 1$, we select additional $\mathcal{B}_{\text{CE}}/\mathcal{R}$ items from $\mathcal{I} \setminus \mathcal{A}_{r-1}$

---

**Algorithm 1** AXN - Test-time $k$-NN Search Inference

---

1: **Input:** $q$: Test query, $V \in \mathbb{R}^{|\mathcal{I}| \times d}$ Item Embeddings, $\mathcal{R}$: Number of iterative search rounds, $k_s$: Number of items to retrieve in each round, $f_\theta$: Cross-Encoder (CE) model
2: **Output:** $\hat{S}$: Approximate scores of $q$ with all items, $\mathcal{A}_\mathcal{R}$: Retrieved items with CE scores in $\mathbf{a}_\mathcal{R}$.
3: $\mathcal{A}_1 \leftarrow \text{INIT}(\mathcal{I}, k_s)$                                                 $\triangleright$ Initial set of items
4: $\mathbf{a}_1 \leftarrow [f_\theta(q,i)]_{i \in \mathcal{A}_1}$                               $\triangleright$ CE scores of $q$ with items in $\mathcal{A}_1$
5: $\mathbf{u}_q \leftarrow \text{Solve-Linear-Regression}(V, \mathcal{A}_1, \mathbf{a}_1)$      $\triangleright$ Compute query embedding by solving Eq.4
6: **for** $r \leftarrow 2$ to $\mathcal{R}$ **do**
7:    $\hat{S}^{(r)} \leftarrow \mathbf{u}_q \times V^\mathsf{T}$                                       $\triangleright$ Update approx. scores
8:    $\mathcal{A}_r \leftarrow \mathcal{A}_{r-1} \cup \arg \text{top} k_{i \in \mathcal{I} \setminus \mathcal{A}_{r-1}, k=k_s} \hat{S}_i^{(r)}$          $\triangleright$ Retrieve $k_s$ new items
9:    $\mathbf{a}_r \leftarrow \mathbf{a}_{r-1} \oplus [f_\theta(q,i)]_{i \in \mathcal{A}_r \setminus \mathcal{A}_{r-1}}$        $\triangleright$ Compute CE scores of new items
10:    $\mathbf{u}_q \leftarrow \text{Solve-Linear-Regression}(V, \mathcal{A}_r, \mathbf{a}_r)$    $\triangleright$ Compute query embedding by solving Eq.4
11: $\hat{S} \leftarrow \mathbf{u}_q \times V^\mathsf{T}$                                         $\triangleright$ Compute approx. scores
12: **return** $\hat{S}, \mathcal{A}_\mathcal{R}, \mathbf{a}_\mathcal{R}$

---

using inner-product of test query embedding $\mathbf{u}_{q_\text{test}}$ and item embeddings $\mathbf{v}_i$ (line 8 in Alg. 1).

$$\mathcal{A}_r = \mathcal{A}_{r-1} \cup \underset{i \in \mathcal{I} \setminus \mathcal{A}_{r-1}, k=\mathcal{B}_\text{CE}/\mathcal{R}}{\arg \text{top} k} \mathbf{u}_{q_\text{test}} \mathbf{v}_i^\mathsf{T} \tag{5}$$

After computing $\mathcal{A}_r$, we compute CE scores for new items chosen in round $r$, and we update the test query embedding $\mathbf{u}_{q_\text{test}}$ by solving Eq. 4 with the latest set of items $\mathcal{A}_r$ which includes additional items selected in round $r$. Note that solving for $\mathbf{u}_{q_\text{test}}$ in Eq 4 is akin to solving a linear regression problem with embeddings of items in $\mathcal{A}_r$ as features and cross-encoder scores of the items as regression targets. We solve Eq. 4 analytically to get $\mathbf{u}_{q_\text{test}} = (V_{\mathcal{A}_r}^\mathsf{T} V_{\mathcal{A}_r})^\dagger V_{\mathcal{A}_r}^\mathsf{T} \mathbf{a}_r$ where $M^\dagger$ denotes pseudo-inverse of a matrix $M$.

At the end of $\mathcal{R}$ rounds, we obtain $\mathcal{A}_\mathcal{R}$ containing $\mathcal{B}_\text{CE}$ items, all of which have been scored using the cross-encoder model. We return top-$k$ items from this set sorted based on exact cross-encoder scores as the set of approximate $k$-NN for given test query $q_\text{test}$

$$\hat{\mathcal{N}}(q_\text{test}) = \underset{i \in \mathcal{A}_\mathcal{R}}{\arg \text{top} k} \, f(q_\text{test}, i) \tag{6}$$

**Regularizing Test Query Embedding**    The system of equation in Eq 4 in round $r$ contains $|\mathcal{A}_r|$ equations with $d$ variables and is an under-determined system when $|\mathcal{A}_r| < d$. In such a case, there exist infinitely many solutions to Eq 4 and the test query embedding $\mathbf{u}_{q_\text{test}}$ can achieve zero approximation error on items in $\mathcal{A}_r$, and may show poor generalization when estimating cross-encoder scores for items in $\mathcal{I} \setminus \mathcal{A}_r$. Since the approximate scores are used to select the additional set of items in round $r+1$ (line 8 in Alg. 1), such poor approximation affects the additional set of items chosen, and subsequently, it may affect the overall retrieval quality in certain settings. To avoid such overfitting, we compute the final test query embedding as:

$$\mathbf{u}_{q_\text{test}} = (1-\lambda)\mathbf{u}_{q_\text{test}}^{(\text{LinReg})} + \lambda \mathbf{u}_{q_\text{test}}^{(\text{param})} \tag{7}$$

where $\mathbf{u}_{q_\text{test}}^{(\text{LinReg})}$ is the analytical solution to the linear system in Eq. 4 and $\mathbf{u}_{q_\text{test}}^{(\text{param})}$ is the test query embedding obtained from a parametric model such as a dual-encoder or an inductive matrix factorization model. We tune the weight parameter $\lambda \in [0, 1]$ on the dev set.

## 3    EXPERIMENTS

In our experiments, we evaluate proposed approaches and baselines on the task of finding $k$-nearest neighbors for cross-encoder (CE) models as well as on downstream tasks. We use cross-encoders trained for the downstream task of zero-shot entity linking and zero-shot information retrieval and present extensive analysis of the effect of various design choices on the offline indexing latency and the test-time retrieval recall.

**Experimental Setup**    We run experiments on two datasets/benchmarks – ZESHEL (Logeswaran et al., 2019), a zero-shot entity linking benchmark, and BEIR benchmark (Thakur et al., 2021b), a collection of information retrieval datasets for evaluating zero-shot performance of IR models. We

use separate CE models for ZeShEL and BeIR datasets, trained using ground-truth labeled data from the corresponding dataset. For evaluation, we use two test domains from ZeShEL dataset –YuGiOh and Star Trek with 10K and 34K items (entities) respectively, and we use SciDocs and Hotpot-QA datasets from BeIR with 25K and 5M items (documents) respectively. These domains were *not* part of the data used to train the corresponding cross-encoder models. Following the precedent set by previous work (Yadav et al., 2022; 2023), we create a train/test split uniformly at random for each ZeShEL domain. For datasets from BeIR, we use pseudo-queries released as part of the benchmark as train queries and test on queries in the official test split in BeIR benchmark. We use queries in the train split to train proposed matrix factorization models or baseline DE models via distillation, and we evaluate on the corresponding domain's test split. We refer interested readers to Appendix A for more details about datasets, cross-encoder training, and model architecture.

**Baselines** We compare with the following retrieve-and-rerank baselines, denoted by $\text{RNR}_X$, where top-scoring items wrt baseline scoring method $X$ are retrieved and then re-ranked using the CE.

- **TF-IDF**: It computes the similarity score for a query-item pair using the dot-product of sparse query/item vectors containing TF-IDF weights.
- **Dual-Encoders (DE)**: It computes query-item scores using the dot-product of dense embeddings produced by encoding queries and items separately. We experiment with two DE models.
  - $\text{DE}_{\text{SRC}}$: DE trained on the same *source* data and downstream task as the cross-encoder model. This model is *not* trained or finetuned on the target domains used for evaluation in this work.
  - $\text{DE}_{\text{DSTL}}$: This corresponds to $\text{DE}_{\text{SRC}}$ further finetuned via distillation using the cross-encoder model on the *target* domain i.e. the domain used for evaluation.

We also compare with AdaCUR (Yadav et al., 2023), a CUR-based approach that computes a dense matrix with CE scores between training queries and all items to index the items, and performs adaptive retrieval at test time. We use $\text{AdaCUR}_X$ to denote inference with AdaCUR method when items in the first round are chosen using method $X \in \{\text{DE}_{\text{SRC}}, \text{TF-IDF}\}$. We refer readers to Appendix A for implementation details for all baselines and proposed approaches.

**Proposed Approach** We construct the sparse matrix $G$ on the target domain by selecting top-scoring items wrt $\text{DE}_{\text{SRC}}$ for each query in $\mathcal{Q}_{\text{train}}$ followed by computing the CE scores for observed query-item pairs in $G$. We use $\text{DE}_{\text{SRC}}$ to initialize embeddings for train queries and all items, followed by inductive ($\text{MF}_{\text{IND}}$) or transductive ($\text{MF}_{\text{TRNS}}$) matrix factorization while minimizing the objective function in 3. We use the same sparse matrix $G$ when training DE via distillation ($\text{DE}_{\text{DSTL}}$) on the target domain. We use $\text{AxN}_{X,Y}$ to denote the proposed $k$-NN search method (§2.2) when using method $X$ to compute item embeddings and method $Y$ to retrieve items in the first round.

**Evaluation Metrics** Following the precedent set by previous work (Yadav et al., 2022; 2023), we use Top-$k$-Recall@$m$ for test queries as the evaluation metric which measures the fraction of $k$-nearest neighbors as per the CE which are present in the set of $m$ retrieved items. For each method, we retrieve $m$ items and re-rank them using exact CE scores. We also evaluate the quality of the retrieved $k$-NN items wrt the CE on the downstream task. We use entity linking accuracy for ZeShEL, and we use downstream task specific nDCG@10 and recall for BeIR domains.

For each approach, we calculate the time taken for indexing a given set of items from the target domain which involves some or all of the following steps: a) computing query/item embeddings using $\text{DE}_{\text{SRC}}$, b) computing (dense or sparse) query-item score matrix $G$ for $\mathcal{Q}_{\text{train}}$, c) gradient-based training using $G$ to estimate item embeddings for $\text{MF}_{\text{TRNS}}$ or parameters of models such as $\text{DE}_{\text{DSTL}}$ and $\text{MF}_{\text{IND}}$, and d) for $\text{DE}_{\text{DSTL}}$ and $\text{MF}_{\text{IND}}$, computing updated item embeddings after training.

## 3.1 Results

Figure 1 shows Top-1-Recall@Inference-Cost=100 and Top-100-Recall@Inference-Cost=500 versus the total wall-clock time taken to index the items for various approaches on YuGiOh and Hotpot-QA. AdaCUR can control the indexing time by varying $|\mathcal{Q}_{\text{train}}|$, the number of train queries, while MF and distillation-based methods can control the indexing time by varying $|\mathcal{Q}_{\text{train}}|$ and the number of items scored per train query ($k_d$). For YuGiOh, we use $|\mathcal{Q}_{\text{train}}| \leq 500$ for all methods, and for Hotpot-QA, we use $|\mathcal{Q}_{\text{train}}| \leq 1\text{K}$ for AdaCUR and $|\mathcal{Q}_{\text{train}}| \leq 50\text{K}$ with other methods.

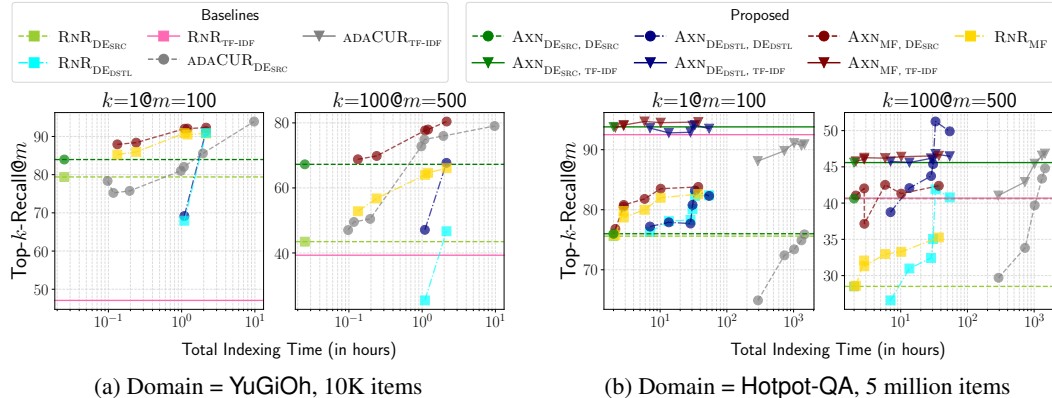

(a) Domain = YuGiOh, 10K items       (b) Domain = Hotpot-QA, 5 million items

Figure 1: Top-1-Recall and Top-100-Recall at inference cost budget ($m$) of 100 and 500 CE calls respectively versus indexing time for various approaches. Matrix factorization approaches (MF) can be significantly faster than ADACUR and training DE via distillation ($DE_{DSTL}$). The proposed adaptive $k$-NN search method (AXN) provides consistent improvement over corresponding retrieve-and-rerank style inference (RNR).

**Proposed Inference (AXN) vs Retrieve-and-Rerank (RNR)**   AXN consistently provides improvement over the corresponding retrieve-and-rerank (RNR) baseline. For instance, $AXN_{DE_{SRC}, DE_{SRC}}$ provides an improvement of 5.2% for $k$=1 and 54% for $k$=100 over $RNR_{DE_{SRC}}$ for domain=YuGiOh. Note that this performance improvement comes at *no additional* offline indexing cost and with negligible test-time overhead[3]. $RNR_{TF-IDF}$ performs poorly on YuGiOh while it serves as a strong baseline for Hotpot-QA, potentially due to differences in task, data, and CE model. On Hotpot-QA, Top-$k$-Recall for AXN can be further improved by sampling items in the first round using TF-IDF ($AXN_{Z,TF-IDF}$) instead of $DE_{SRC}$ ($AXN_{Z,DE_{SRC}}$) for all indexing methods $Z \in \{DE_{SRC}, DE_{DSTL}, MF\}$.

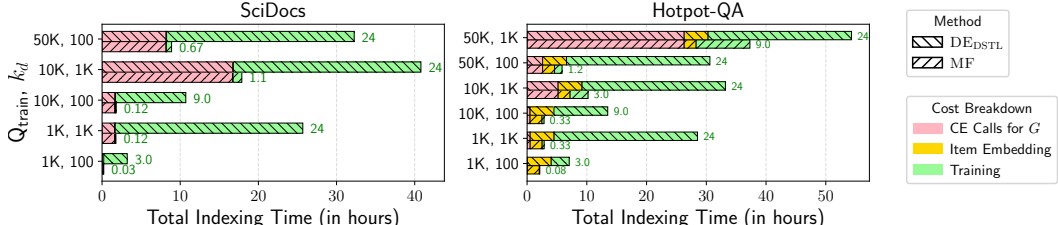

Figure 2: Breakdown of indexing latency of MF and $DE_{DSTL}$ into various steps with training time shown on the right of each bar for different values of $|\mathcal{Q}_{train}|$ and no. of items scored per query ($k_d$).

**Matrix Factorization vs $DE_{DSTL}$**   Unsurprisingly, performance on the target domain can be further improved by using data from the target domain to fit an embedding space to approximate the CE. As shown in Figure 1, our proposed matrix factorization based approaches (MF) can be significantly more efficient than the distillation-based ($DE_{DSTL}$) approaches while matching or outperforming $DE_{DSTL}$ in terms of $k$-NN search recall in the majority of the cases. Figure 2 shows the breakdown of total indexing time of $DE_{DSTL}$ and MF for different numbers of training queries ($|\mathcal{Q}_{train}|$) and number of items scored per query ($k_d$) using the CE in the sparse matrix $G$. As expected, both the time taken to compute $G$ and the training time increases with the number of queries and the number of items scored per query. The training time does not increase proportionally after 10K queries as we allocated a maximum training time of 24 hours for all methods. For MF, the majority of the time is spent either in computing sparse matrix $G$ or the initial item embeddings. While we report total GPU hours taken for CE calls to compute $G$ and initial item embeddings, these steps can be easily parallelized across multiple GPUs without any communication overhead. Since $DE_{DSTL}$ trains all parameters of a large parametric neural model, it requires large amounts of GPU memory and takes up to several hours [4]. In contrast, MF-approaches require significantly less memory[5] and training time as these approaches train the item embeddings as free parameters ($MF_{TRNS}$) or train a

---

[3]We refer readers to §B.1 for analysis of overhead incurred by AXN

[4]We trained dual-encoders on an Nvidia RTX8000 GPU with 48 GB memory for a maximum of 24 hours.

[5]We used an Nvidia 2080ti with 12 GB memory for MF-based methods.

shallow neural network on top of fixed embeddings ($\text{MF}_{\text{IND}}$) from an existing DE. We report results for $\text{MF}_{\text{TRNS}}$ on small-scale domains (e.g. YuGiOh with 10K items) and for $\text{MF}_{\text{IND}}$ on large-scale domain Hotpot-QA (5 million items). We refer interested readers to Appendix B.3 for comparison of $\text{MF}_{\text{TRNS}}$ and $\text{MF}_{\text{IND}}$ on small- and large-scale datasets.

**Proposed Approaches vs ADACUR** Our proposed inference method (AXN) in combination with MF or DE can outperform or closely match the performance of ADACUR while requiring orders of magnitude less compute for the offline indexing stage, on both small- and large-scale datasets. For instance, $\text{ADACUR}_{\text{DE}_{\text{SRC}}}$ requires 1000+ GPU hours for embedding 5 million items in Hotpot-QA, and achieves Top-1-Recall@100 = 75.9 and Top-100-Recall@500 = 44.8. In contrast, $\text{MF}_{\text{IND}}$ with $|\mathcal{Q}_{\text{train}}|$=10K and 100 items per query takes less than three hours to fit item embeddings, and $\text{AXN}_{\text{MF}_{\text{IND}},\text{DE}_{\text{SRC}}}$ achieves Top-1-Recall@100 = 80.5 and Top-100-Recall@500 = 42.6.

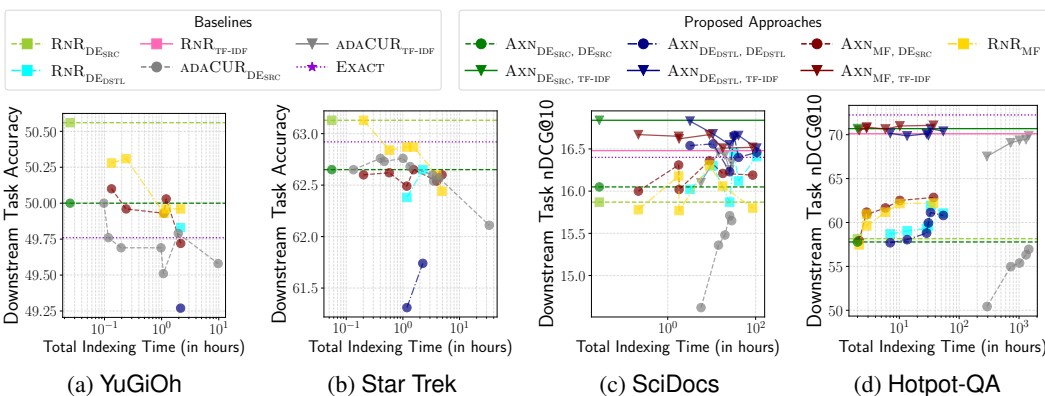

Figure 3: Downstream task performance versus indexing time for proposed and baseline approaches on different domains. All methods use a fixed inference cost budget of 100 cross-encoder calls.

**Downstream Task Performance** Figure 3 shows downstream task performance on proposed and baseline approaches including EXACT which performs exact brute-force search using CE at test-time. For Hotpot-QA, we observe that improvement in $k$-NN search accuracy results in improvement in downstream task performance with EXACT brute-force performing the best. We observe a different trend on SciDocs, YuGiOh, and Star Trek where EXACT search results in suboptimal performance as compared to $\text{RNR}_{\text{DE}_{\text{SRC}}}$. For instance, $\text{RNR}_{\text{DE}_{\text{SRC}}}$ achieves accuracy of 50.6 while the accuracy of EXACT is 49.8 on the downstream task of entity linking on YuGiOh. We believe that this difference in trends in $k$-NN search performance and downstream task performance could be due to differences in the training setup of the corresponding CE (i.e. the loss function and negatives used during training, see Appendix A.1 for details) as well as the nature of the task and data. While beyond the scope of this paper, it would be interesting to explore different loss functions and training strategies such as using negative items mined using $k$-NN search strategies proposed in this work to improve the robustness and generalization capabilities of cross-encoders and minimize such discrepancies in $k$-NN search and downstream task performance.

We refer readers to Appendix B for an analysis of the overhead incurred by AXN (§B.1), a comparison of AXN with pseudo-relevance feedback based approaches (§B.2), an analysis of design choices for our proposed approach (§B.3,B.4), and results on other downstream evaluation metrics for BEIR.

## 4 RELATED WORK

**Approximating Similarity Function** Matrix factorization methods have been widely used for computing low-rank approximation of dense distance and kernel matrices (Musco & Woodruff, 2017; Bakshi & Woodruff, 2018; Indyk et al., 2019), non-PSD matrices (Ray et al., 2022) as well as for estimating missing entries in sparse matrices (Koren et al., 2009; Luo et al., 2014; Yu et al., 2014; Mehta & Rana, 2017; Xue et al., 2017). In this work, we focus on methods for factorizing sparse matrices instead of dense matrices as computing each entry in the matrix (i.e. CE score for a query-item pair) requires a forward-pass through an expensive neural model. An essential assumption for matrix completion methods is that the underlying matrix $M$ is low-rank, thus enabling recovery of the missing entries while only observing a small fraction of entries in $M$ (Candes & Recht, 2012;

Nguyen et al., 2019). Theoretically, such matrix completion methods require $\Omega(nr)$ samples to recover an $m \times n$ matrix of rank $r$ with $m \leq n$ (Krishnamurthy & Singh, 2013; Xu et al., 2015). The sample complexity can be improved in the presence of features describing rows and columns of the matrix, often referred to as side information (Jain & Dhillon, 2013; Xu et al., 2013; Zhong et al., 2019). Inductive matrix completion ($MF_{IND}$) approaches leverage such query and item features to improve the sample complexity and also enable generalization to unseen queries (rows) and items (columns). Training dual-encoder (DE) models via distillation using a cross-encoder (CE), where the DE consumes raw query and item features (such as query/item description) and produces query/item embeddings, can be seen as solving an inductive matrix factorization problem. A typical training objective for training DE involves minimizing the discrepancy between CE (teacher model) and DE (student model) scores on observed entries in the sparse matrix (Hofstätter et al., 2020; Reddi et al., 2021; Thakur et al., 2021a). Recent work has explored different strategies for distillation-based training of DE such as curriculum learning based methods (Zeng et al., 2022), joint training of CE and DE to mutually improve the performance of both models (Liu et al., 2022; Ren et al., 2021). Inductive MF methods ($MF_{IND}$) used in this work also share similar motivations to adapters (Houlsby et al., 2019) which introduce a small number of trainable parameters between layers of the model, and may reduce training time and memory requirements in certain settings (Rücklé et al., 2021). $MF_{IND}$ used in this work only trains a shallow MLP on top of query/item embeddings from DE while keeping DE parameters frozen, and does not introduce any parameters in the DE.

**Nearest Neighbor Search**    $k$-NN search has been widely studied in applications where the inputs are described as vectors in $\mathbb{R}^d$ (Clarkson et al., 2006; Li et al., 2019), and the similarity is computed using simple (dis-)similarity functions such as inner-product (Johnson et al., 2019; Guo et al., 2020) and $\ell_2$-distance (Kleinberg, 1997; Chávez et al., 2001; Hjaltason & Samet, 2003). These approaches typically work by speeding up each distance/similarity computation (Jegou et al., 2010; Hwang et al., 2012; Zhang et al., 2014; Yu et al., 2017; Bagaria et al., 2021) as well as constructing tree-based (Beygelzimer et al., 2006; Dong et al., 2020) or graph-based data structures (Malkov & Yashunin, 2018; Wang et al., 2021a; Groh et al., 2022) over the given item set to efficiently navigate and prune the search space to find (approximate) $k$-NN items for a given query. Recent work also explores such graph-based (Boytsov & Nyberg, 2019a; Tan et al., 2020; 2021; MacAvaney et al., 2022), or tree-based (Boytsov & Nyberg, 2019b) data structures for non-metric and parametric similarity functions. Another line of work explores model quantization (Nayak et al., 2019; Liu et al., 2021) and early-exit strategies (Xin et al., 2020a;b) to approximate the neural model while speeding up each forward pass through the model and reducing its memory footprint. It would be interesting to study if such data structures and approaches to speed up cross-encoder score computation can be combined with matrix factorization based approaches proposed in this work to further improve recall-vs-cost trade-offs for $k$-NN search with cross-encoders.

**Pseudo-Relevance Feedback (PRF)**    Similar to PRF-based methods in information retrieval (Rocchio Jr, 1971; Lavrenko & Croft, 2001), our proposed $k$-NN search method AXN refines the test query representation using model-based feedback. In our case, we use the cross-encoder scores of items retrieved in the previous round as feedback to update the test query representation. PRF-based approaches have been widely used in information retrieval for retrieval with sparse(Li et al., 2018; Mao et al., 2020; 2021) and dense embeddings (Yu et al., 2021; Wang et al., 2021b). We refer readers to Appendix §B.2 for comparison with a recent PRF-based method (Sung et al., 2023).

## 5    CONCLUSION

In this paper, we present an approach to perform $k$-NN search with cross-encoders by efficiently approximating the cross-encoder scores using dot-product of learned test query and item embeddings. In the offline indexing step, we compute item embeddings to index a given set of items from a target domain by factorizing a sparse query-item score matrix, leveraging existing dual-encoder models to initialize the item embeddings while avoiding computationally-expensive distillation-based training of dual-encoder models. At test time, we compute the test query embedding to approximate cross-encoder scores of the given test query for a small set of adaptively-chosen items, and perform retrieval with the approximate cross-encoder scores. We perform extensive empirical analysis on two zero-shot retrieval benchmarks and show that our proposed approach provides significant improvement in test-time $k$-NN search recall-vs-cost tradeoffs while still requiring significantly less compute resources for indexing items from a target domain as compared to previous approaches.

## ACKNOWLEDGMENTS

We thank members of UMass IESL for helpful discussions and feedback. This work was supported in part by the Center for Data Science and the Center for Intelligent Information Retrieval, in part by the National Science Foundation under Grant No. NSF1763618, in part by the Chan Zuckerberg Initiative under the project "Scientific Knowledge Base Construction", in part by International Business Machines Corporation Cognitive Horizons Network agreement number W1668553, in part by Amazon Digital Services, and in part using highperformance computing equipment obtained under a grant from the Collaborative R&D Fund managed by the Massachusetts Technology Collaborative. Any opinions, findings, conclusions, and recommendations expressed in this material are those of the authors and do not necessarily reflect those of the sponsor(s).

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

## A  Training and Implementation Details

| Dataset | Domain | $|\mathcal{I}|$ | $(|\mathcal{Q}_{\text{train}}|/|\mathcal{Q}_{\text{test}}|)$ Splits | Train Query ($\mathcal{Q}_{\text{train}}$) Type |
|---------|--------|------|----------------------------------|------------------------|
| ZESHEL | YuGiOh | 10,031 | (100/3274), (500/2874), (2000/1374) | Real Queries |
| ZESHEL | Star Trek | 34,430 | (100/4127), (500/3727), (2000/2227) | Real Queries |
| BEIR | SciDocs | 25,657 | {1K, 10K, 50K}/1000 | Pseudo-Queries |
| BEIR | Hotpot-QA | 5,233,329 | {1K, 10K, 50K}/1000 | Pseudo-Queries |

Table 1: Statistics on number of items ($\mathcal{I}$), number of queries in train ($\mathcal{Q}_{\text{train}}$) and test ($\mathcal{Q}_{\text{test}}$) splits for each domain. Following the precedent set by Yadav et al. (2022), we create train/test split by splitting the queries in each ZESHEL domain uniformly at random, and experiment with three values of $|\mathcal{Q}_{\text{train}}| \in \{100, 500, 2000\}$. For BEIR domains, we use pseudo-queries released as part of the benchmark as train queries ($\mathcal{Q}_{\text{train}}$) and run $k$-NN evaluation on test-queries from the official test-split (as per BEIR benchmark) of these domains. For HotpotQA, we use the first 1K queries out of a total of 7K test queries and we use all 1K test queries for SciDocs.

### A.1  Training Cross-Encoder Models

In our experiments, we use [EMB]-CE, a cross-encoder model variant proposed by Yadav et al. (2022) that jointly encodes a query-item pair and computes the final score using dot-product of contextualized query and item embeddings extracted after joint encoding.

**ZESHEL Dataset**  For ZESHEL, we use the cross-encoder model checkpoint[6] released by Yadav et al. (2022). We refer readers to Yadav et al. (2022) for further details on parameterization and training of the cross-encoder.

---

[6] https://huggingface.co/nishantyadav/emb_crossenc_zeshel

**BEIR Benchmark** For BEIR, we use the cross-encoder model checkpoint[7] trained on MS-MARCO dataset and released by Yadav et al. (2023). The cross-encoder model is parameterized using a 6-layer MINI-LM[8] model (Wang et al., 2020) and uses the dot-product based scoring mechanism for cross-encoders proposed by Yadav et al. (2022).

## A.2 TRAINING DUAL-ENCODER AND MATRIX FACTORIZATION MODELS

For BEIR datasets, we train matrix factorization models and $\text{DE}_{\text{DSTL}}$ using sparse matrix $G$ containing number of train queries $|\mathcal{Q}_{\text{train}}| \in \{1K, 10K, 50K\}$ with number of items per query $k_d \in \{100, 1000\}$. For ZESHEL datasets, we use $|\mathcal{Q}_{\text{train}}| \in \{100, 500, 2000\}$ with the number of items per query $k_d \in \{100, 1000\}$ for matrix factorization models and $k_d \in \{25, 100\}$ for training $\text{DE}_{\text{DSTL}}$ model. Table 1 shows train/test splits used for each domain.

### A.2.1 TRAINING DUAL-ENCODER MODELS

We train dual-encoder models on Nvidia RTX8000 GPUs with 48 GB GPU memory.

**ZESHEL dataset** We report results for DE baselines as reported in Yadav et al. (2022). The DE models were initialized using `bert-base-uncased` and contain separate query and item encoders, thus resulting in a total of $2 \times 110M$ parameters. The DE models are trained using cross-entropy loss to match the DE score distribution with the CE score distribution. We refer readers to Yadav et al. (2022) for details related to training of DE models on ZESHEL dataset.

**BEIR benchmark** For BEIR domains, we use a dual-encoder model checkpoint[9] released as part of `sentence-transformer` repository as $\text{DE}_{\text{SRC}}$, unless specified otherwise. This DE model was initialized using `distillbert-base` (Sanh et al., 2019) model and trained on MS-MARCO dataset which contains 40 million (query, positive document (item), negative document (item)) triplets using triplet ranking loss. This $\text{DE}_{\text{SRC}}$ is not trained on target domains **SciDocs** and **Hotpot-QA** used for running $k$-NN experiments in this paper. We finetune $\text{DE}_{\text{SRC}}$ via distillation on the target domain to get the $\text{DE}_{\text{DSTL}}$ model. Given a set of training queries $\mathcal{Q}_{\text{train}}$ from the target domain, we retrieve top-100 or top-1000 items for each query, score the items with the cross-encoder model and train the dual-encoder by minimizing cross-entropy loss between predicted query-item scores (using DE) and target query-item scores (obtained using CE). We train $\text{DE}_{\text{DSTL}}$ using AdamW (Loshchilov & Hutter, 2019) optimizer with learning rate 1e-5 and accumulating gradient over 4 steps. We trained for 10 epochs when using top-100 items per query and for 4 epochs when using top-1000 items per query. We allocate a maximum time of 24 hours for training.

### A.2.2 MATRIX-FACTORIZATION MODELS

We train both transductive ($\text{MF}_{\text{TRNS}}$) and inductive ($\text{MF}_{\text{IND}}$) matrix factorization models on Nvidia 2080ti GPUs with 12 GB GPU memory for all datasets with the exception that we trained $\text{MF}_{\text{TRNS}}$ for **Hotpot-QA** on Nvidia A100 GPUs with 80 GB GPU memory. We use AdamW optimizer (Loshchilov & Hutter, 2019) with learning rate and number of epochs as shown in Table 2. Training $\text{MF}_{\text{TRNS}}$ on **Hotpot-QA** required 80 GB GPU memory as it involved training 768-dimensional embeddings for 5 million items which roughly translates to around 4 billion trainable parameters, and we used AdamW optimizer with stores additional memory for each trainable parameter. For smaller datasets with the number of items of the order of 50K, smaller GPUs with 12 GB memory sufficed.

For inductive matrix factorization ($\text{MF}_{\text{IND}}$), we train a 2-layer MLP with skip-connection on top of query and item embeddings from $\text{DE}_{\text{SRC}}$. For a given input embedding $x_{\text{in}} \in \mathbb{R}^d$, we compute the output embedding $x_{\text{out}} \in \mathbb{R}^d$ as

$$x'_{\text{out}} = b_2 + W_2^\intercal \text{gelu}(b_1 + W_1^\intercal x_{\text{in}})$$
$$x_{\text{out}} = \sigma(w_{\text{skip}})x'_{\text{out}} + (1 - \sigma(w_{\text{skip}}))x$$

---

[7] https://huggingface.co/nishantyadav/emb_crossenc_msmarco_miniLM

[8] https://huggingface.co/sentence-transformers/all-MiniLM-L6-v2

[9] `msmarco-distilroberta-base-v2`: www.sbert.net/docs/pretrained-models/msmarco-v2.html

where $W_1 \in \mathbb{R}^{d \times 2d}, b_1 \in \mathbb{R}^{2d}, W_2 \in \mathbb{R}^{2d \times d}, b_2 \in \mathbb{R}^d, w_{\text{skip}} \in \mathbb{R}$ are learnable parameters and $\sigma(.)$ is the sigmoid function. We initialize $w_{\text{skip}}$ with -5 and use default PyTorch initialization for other parameters. We trained separate MLP models for queries and items. We would like to highlight that a simple 2-layer MLP *without* the skip connection i.e. using $x'_{\text{out}}$ as the final output embedding performed poorly in our experiments and it did not generalize well to unseen queries and items.

| Domain | MF Type | Learning Rate | Number of Epochs |
|--------|---------|---------------|------------------|
| SciDocs | $\text{MF}_{\text{TRNS}}$ | 0.005 | 4 if $(|\mathcal{Q}_{\text{train}}|, k_d) \in \{(10\text{K},1\text{K}), (50\text{K}, 1\text{K}\}$ else 10 |
| SciDocs | $\text{MF}_{\text{IND}}$ | 0.005 | 10 if $(|\mathcal{Q}_{\text{train}}|, k_d) \in \{(10\text{K},1\text{K}), (50\text{K}, 1\text{K}\}$ else 20 |
| Hotpot-QA | $\text{MF}_{\text{TRNS}}$ | 0.001 | 4 if $(|\mathcal{Q}_{\text{train}}|, k_d) \in \{(10\text{K},1\text{K}), (50\text{K}, 1\text{K}\}$ else 10 |
| Hotpot-QA | $\text{MF}_{\text{IND}}$ | 0.001 | 10 if $(|\mathcal{Q}_{\text{train}}|, k_d) \in \{(10\text{K},1\text{K}), (50\text{K}, 1\text{K}\}$ else 20 |
| YuGiOh | $\text{MF}_{\text{TRNS}}$ | 0.001 | 20 |
| Star Trek | $\text{MF}_{\text{TRNS}}$ | 0.001 | 20 |

Table 2: Hyperparameters for transductive ($\text{MF}_{\text{TRNS}}$) and inductive ($\text{MF}_{\text{IND}}$) matrix factorization models for different number of training queries ($|\mathcal{Q}_{\text{train}}|$) and number of items per train query ($k_d$) in sparse matrix $G$.

### A.3 TF-IDF

For BEIR datasets, we use BM25 with parameters as reported in Thakur et al. (2021b) and for ZESHEL, we use TF-IDF with default parameters from Scikit-learn (Pedregosa et al., 2011), as reported in Yadav et al. (2022).

### A.4 TEST-TIME INFERENCE WITH AXN, ADACUR, AND RNR

For $\text{RNR}_X$, we retrieve top-scoring items using dot-product of query and item embeddings computed using baseline retrieval method $X$ and re-rank the retrieved items using the cross-encoder model. For $\text{RNR}_{\text{MF}_{\text{TRNS}}}$, we use dense query embedding from base dual-encoder model $\text{DE}_{\text{SRC}}$ for test-queries $q_{\text{test}} \notin \mathcal{Q}_{\text{train}}$ along with item embeddings learnt using transductive matrix factorization to retrieve-and-rerank items for the given test query.

For both ADACUR and AXN, we use $\mathcal{R} = 10$ for domains in BEIR and $\mathcal{R} = 5$ for domains in ZESHEL unless stated otherwise. For BEIR datasets, we tune AXN weight parameter $\lambda$ (in eq 7) on the dev set. We refer interested readers to §B.2 for the effect of $\lambda$ on final performance. For ZESHEL, we report results for $\lambda = 0$. For Hotpot-QA, we restrict our $k$-NN search with $\text{AXN}_{X,Y}$ and $\text{ADACUR}_Y$ to top-10K items wrt method $Y$, $Y \in \{\text{DE}_{\text{SRC}}, \text{TF-IDF}\}$. For other domains, we do not use any such heuristic and search over all items.

**Cross-Encoder Score Normalization for AXN** Figure 4a shows query-item score distribution for the cross-encoder model and $\text{DE}_{\text{SRC}}$ on SciDocs datasets from BEIR benchmark. For cross-encoder models trained on BEIR dataset, we observe that the cross-encoder and $\text{DE}_{\text{SRC}}$ model produce query-item scores in significantly different ranges. Since $\text{DE}_{\text{SRC}}$ is used to initialize the embedding space for matrix factorization approaches, this resulted in a mismatch in the range of the target score distribution from the cross-encoder in sparse matrix $G$ and the initial predicted score distribution from $\text{DE}_{\text{SRC}}$. Consequently, using raw cross-encoder scores while training MF models and while computing test query embedding by solving the linear regression problem in Eq 4 leads to a poor approximation of the cross-encoder. To alleviate this issue, we normalize the cross-encoder scores to match the score distribution from $\text{DE}_{\text{SRC}}$ model using two parameters $\alpha, \beta \in \mathbb{R}$.

$$s_{\text{final}}(q, i) = \beta(s_{\text{init}}(q, i) - \alpha)$$

where $s_{\text{init}}(q, i)$ and $s_{\text{final}}(q, i)$ are initial and normalized cross-encoder scores, and $\alpha$ and $\beta$ are estimated by re-normalizing cross-encoder distribution to match dual-encoder score distribution using 100 training queries. Note that such score normalization does not affect the final ranking of items.

We do *not* perform any such normalization for ZESHEL datasets the cross-encoder and $\text{DE}_{\text{SRC}}$ model output scores in similar ranges as shown in Figure 4b.

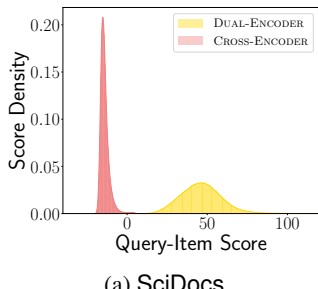

(a) SciDocs

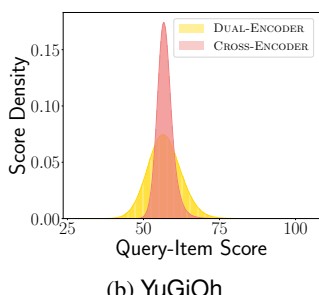

(b) YuGiOh

Figure 4: Score distribution for cross-encoder (CE) and dual-encoder (DE) models on SciDocs for BEIR and YuGiOh from ZESHEL. For each domain, we use cross-encoder and dual-encoder models trained on the corresponding task. See §A.1 for details on cross-encoder training and §A.2.1 for dual-encoder training.

## B  ADDITIONAL RESULTS AND ANALYSIS

### B.1  OVERHEAD OF ADAPTIVE RETRIEVAL WITH AXN

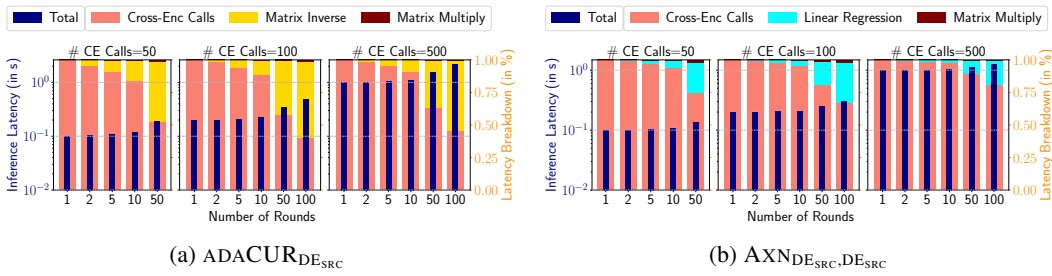

(a) $\text{ADACUR}_{\text{DE}_{\text{SRC}}}$

(b) $\text{AXN}_{\text{DE}_{\text{SRC}}, \text{DE}_{\text{SRC}}}$

Figure 5: Breakdown of inference latency for $\text{ADACUR}_{\text{DE}_{\text{SRC}}}$ and $\text{AXN}_{\text{DE}_{\text{SRC}}, \text{DE}_{\text{SRC}}}$ under different test-time CE call budgets for domain=Hotpot-QA. See §B.1 for detailed discussion.

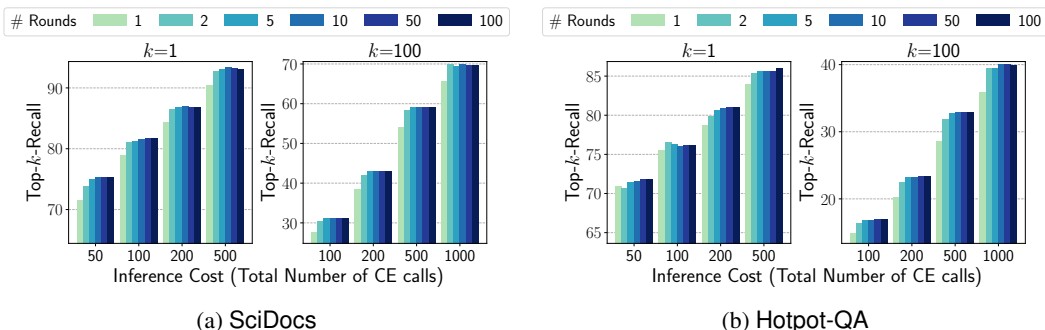

(a) SciDocs

(b) Hotpot-QA

Figure 6: Top-$k$-Recall versus number of rounds for $\text{AXN}_{\text{DE}_{\text{SRC}}, \text{DE}_{\text{SRC}}}$ under different test-time cross-encoder call budgets for domains Hotpot-QA and SciDocs. Number of rounds ($\mathcal{R}$) = 1 corresponds to retrieve-and-rerank style inference with $\text{DE}_{\text{SRC}}$ i.e. $\text{RNR}_{\text{DE}_{\text{SRC}}}$. Top-$k$-Recall generally improves with the number of rounds and saturates around 5 to 10 rounds.

Figures 5a and 5b show total inference latency for $\text{ADACUR}$ and $\text{AXN}$ for varying number of rounds ($\mathcal{R}$) at different cross-encoder (CE) calls budgets. The secondary y-axis in Figure 5 shows the breakdown of the inference latency into three main steps in Algorithm 1 - (a) CE Calls: computing CE scores for retrieved items (line 9), (b) solving linear regression problem to update test query embedding for $\text{AXN}$ (line 10) (c) Matrix Multiply: updating approximate scores for all items (line 7) followed by retrieving items using approximate scores. In case of $\text{ADACUR}$, computing query embedding in step (b) involves computing the pseudo-inverse of a matrix instead of solving a linear regression problem.

As shown in Figure 5, the overhead of adaptive retrieval is negligible for $\mathcal{R} = 5$ to $10$, and the overhead increases linearly with the number of rounds. $\text{AXN}_{\text{DE}_{\text{SRC}},\text{DE}_{\text{SRC}}}$ for $\mathcal{R} = 1$ corresponds to $\text{RNR}_{\text{DE}_{\text{SRC}}}$, retrieve-and-rerank style inference using $\text{DE}_{\text{SRC}}$. We observe that AXN incurs less overhead than ADACUR under the same test-time CE call budget. Each CE call takes an amortized time of $\sim 2\,\text{ms}$[10] when computing CE scores with a batch-size of up to 50 for domain=Hotpot-QA. While the time complexity of updating the approximate scores is linear in the number of items, we observe that this step can be significantly sped up using GPUs/TPUs, and use of efficient vector-based $k$-NN search methods. In this work, to get an efficient implementation for large domains such as Hotpot-QA, we first shortlist 10K items for the test query using the baseline retrieval method (e.g. $\text{DE}_{\text{SRC}}$), and only update the approximate scores for those 10K during inference using brute-force computation of scores for all 10K items. Further, note that the approximate scores are only used for retrieving items (line 8 in Alg. 1), and this operation can also be implemented on CPUs using efficient vector-based $k$-NN search methods (Malkov & Yashunin, 2018; Guo et al., 2020) without the need for brute-force computation of approximate scores for all items.

## B.2 COMPARING DIFFERENT QUERY EMBEDDING METHODS

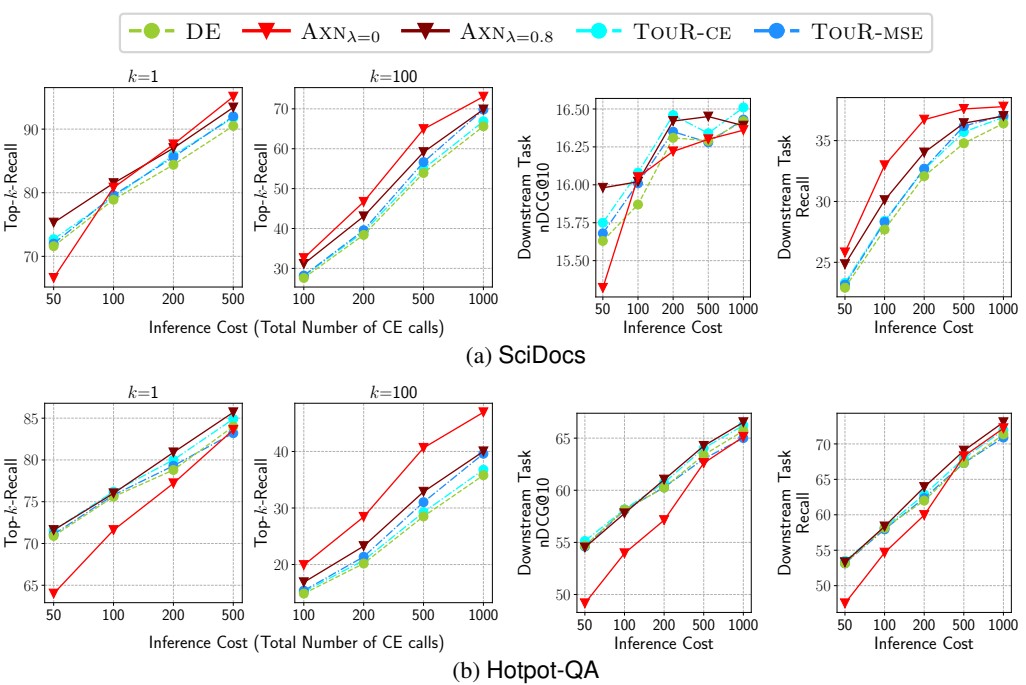

Figure 7: Top-$k$-Recall versus inference cost for different test query embedding methods on domains SciDocs and Hotpot-QA. See §B.2 for detailed discussion.

Our proposed $k$-NN search method shares a similar motivation to pseudo-relevance feedback (PRF) methods that aim to improve the quality of retrieval by updating the initial query representation using heuristic or model-based feedback on retrieved items. We show results for TOUR (Sung et al., 2023), a recent PRF-based method that, similar to our method, also optimizes the test query representations using retrieval results while utilizing the CE call budget of $\mathcal{B}_{\text{CE}}$ CE calls over $\mathcal{R}$ rounds. However, unlike AXN, TOUR uses a single gradient-based update to query embedding to minimize KL-Divergence (TOUR-CE) or mean-squared error (TOUR-MSE) between approximate and exact scores for top-$\mathcal{B}_{\text{CE}}/\mathcal{R}$ items in each round. In contrast, AXN computes the analytical solution to the least-square problem in Eq. 4 in each round, and optionally computes a weighted sum with the test query embedding from a dense parametric model such as a dual-encoder using weight $\lambda \in [0, 1]$ in Eq. 7. For TOUR-CE, we use learning rate $=0.1$ (chosen from $\{0.1, 0.5, 1.0\}$) and for TOUR-MSE, we use learning rate $= 1\text{e-}3$ (chosen from $\{1\text{e-}2, 1\text{e-}3, 1\text{e-}4\}$).

---

[10]On an Nvidia 2080ti GPU with 12 GB memory for a 6-layer Mini-LM (Wang et al., 2020) based model.

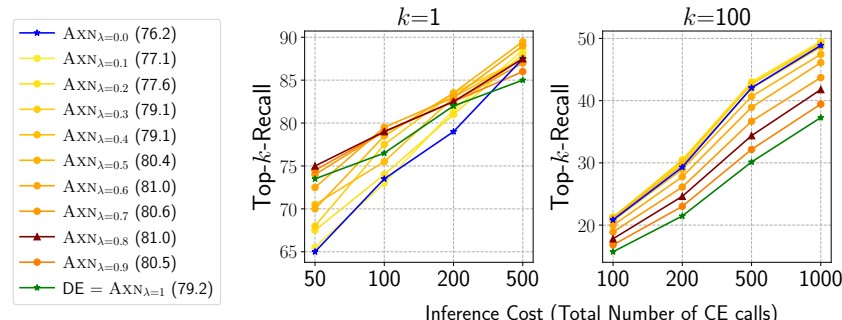

Figure 8: Top-$k$-Recall for $\text{AXN}_{\text{DE}_{\text{SRC}}, \text{DE}_{\text{SRC}}}$ for different values of $\lambda$ parameter in eq 7. We use 200 queries from the validation set in Hotpot-QA and the value in parentheses in the legend denotes average Top-1-Recall, averaged over different test-time inference cost budgets. For $k = 1$, using $\lambda = 0.8$ yields the best performance and for $k = 100$, we use $\lambda = 0$ unless specified otherwise.

Figure 7 shows Top-$k$-Recall and downstream task metrics versus test-time inference CE cost budget ($\mathcal{B}_{\text{CE}}$) for $\text{AXN}_{\text{DE}_{\text{SRC}}, \text{DE}_{\text{SRC}}}$ under two settings of the weight parameter, $\lambda = 0$ and $0.8$, and for $\text{DE}_{\text{SRC}}$ and TOUR baselines. For both SciDocs and Hotpot-QA, $\text{AXN}_{\lambda=0.8}$ performs better than $\text{AXN}_{\lambda=0}$ for $k$-NN search when $k = 1$ while $\lambda = 0$ works better for searching for $k$=100 nearest neighbors. TOUR and AXN achieve similar Top-1-Recall at smaller inference costs with AXN performing marginally better than TOUR at larger cost budgets. However, for $k = 100$, $\text{AXN}_{\lambda=0}$ achieves significantly better recall than TOUR. We observe mixed trends for downstream task metrics. For instance, $\text{AXN}_{\lambda=0.8}$ and TOUR baselines yield similar performance for nDCG@10 on both SciDocs and Hotpot-QA and for downstream task recall on Hotpot-QA while $\text{AXN}_{\lambda=0}$ performs better than all baselines on downstream task recall for SciDocs.

## B.3 TRANSDUCTIVE VERSUS INDUCTIVE MATRIX FACTORIZATION

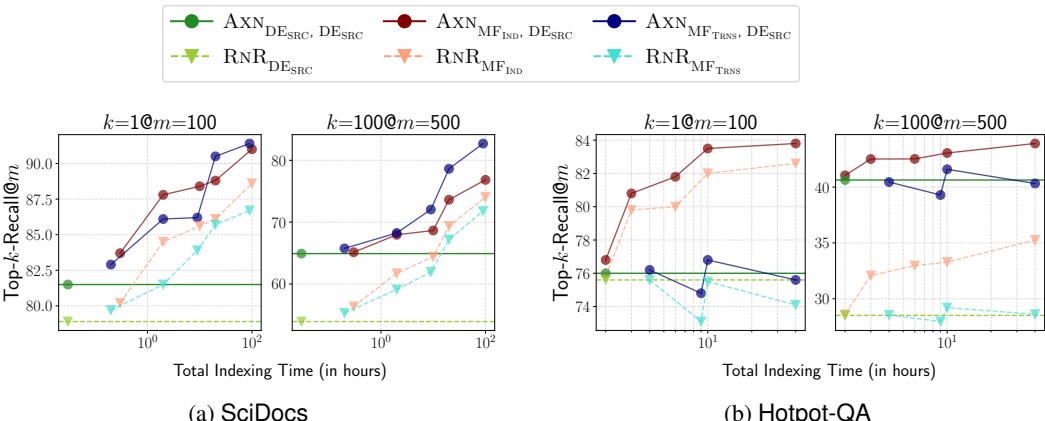

Figure 9: Top-$k$-Recall versus indexing time for transductive ($\text{MF}_{\text{TRNS}}$) and inductive ($\text{MF}_{\text{IND}}$) matrix factorization for SciDocs and Hotpot-QA. We report Top-1-Recall and Top-100-Recall at fixed inference cost budget ($m$) of 100 and 500 CE calls respectively. See §B.3 for detailed discussion.

Figure 9 shows Top-$k$-Recall versus indexing time for $\text{DE}_{\text{SRC}}$, and transductive ($\text{MF}_{\text{TRNS}}$) and inductive ($\text{MF}_{\text{IND}}$) matrix factorization in combination with two test-time inference methods: proposed inference method (AXN) and retrieve-and-rerank style (RNR) inference. We construct the sparse matrix $G$ by selecting top-$k_d$ items for each train query using $\text{DE}_{\text{SRC}}$, and report results for $|\mathcal{Q}_{\text{train}}| \in \{1K, 10K, 50K\}$ and $k_d \in \{100, 1000\}$. We use $\text{DE}_{\text{SRC}}$ to initialize the query and item embeddings for MF methods.

Recall that $\text{MF}_{\text{TRNS}}$ trains item embeddings as free-parameters, and thus requires scoring an item against a small number of train queries in order to update the item embedding. For this reason, $\text{MF}_{\text{TRNS}}$ performs marginally better than or at par with $\text{MF}_{\text{IND}}$ on small-scale data SciDocs with 25K items, as selecting even for $|\mathcal{Q}_{\text{train}}| = 1000, k_d = 100$, results in each item being scored with

four queries on average. However, $\text{MF}_{\text{TRNS}}$ performs poorly for large-scale data Hotpot-QA (with 5 million items) due to the increased sparsity of matrix $G$, providing marginal to no improvement over $\text{DE}_{\text{SRC}}$. In contrast, $\text{MF}_{\text{IND}}$ provides consistent improvement over $\text{DE}_{\text{SRC}}$ on Hotpot-QA.

## B.4 Effect of Sparse Matrix Construction Strategy

| Sparse Matrix Construction Strategy | $\|\mathcal{Q}_{\text{train}}\|, k_d$ | Hotpot-QA | | | SciDocs | | |
|---|---|---|---|---|---|---|---|
| | | Time to compute $G$ | Train-Time $\text{MF}_{\text{IND}}$ | Train-Time $\text{MF}_{\text{TRNS}}$ | Time to compute $G$ | Train-Time $\text{MF}_{\text{IND}}$ | Train-Time $\text{MF}_{\text{TRNS}}$ |
| $k_d$ items per query | 1K, 100 | 3 mins | 5 mins (20) | - | 10 mins | 5 mins (10) | 1.5 mins (10) |
| | 1K, 1000 | 31 mins | 20 mins (20) | - | 1.6 hrs | 20 mins (20) | 7 mins (10) |
| | 10K, 100 | 30 mins | 20 mins (20) | 1.2 hrs (10) | 1.6 hrs | 20 mins (20) | 7.5 mins (10) |
| | 10K, 1000 | 5.2 hrs | 3 hrs (20) | 3.2 hrs ( 4 ) | 16.7 hrs | 3.2 hrs (20) | 1.1 hrs ( 4) |
| | 50K, 100 | 2.6 hrs | 1.2 hrs (20) | 4.1 hrs (10) | 8.3 hrs | 1.3 hrs (20) | 0.6 hrs (10) |
| | 50K, 1000 | 26.3 hrs | 9 hrs (10) | 16 hrs ( 4) | 82 hrs | 14 hrs (10) | 3.7 hrs ( 4) |
| $k_d$ queries per item | 50K, 2 | 5.8 hrs | 3hrs (20) | 7.5 hrs (10) | 5 mins | 3 mins (20) | 6.5 mins (20) |
| | 50K, 5 | 12.7 hrs | 8hrs (20) | 8.5 hrs ( 4) | 14 mins | 5 mins (20) | 9 mins (20) |
| | 50K, 10 | 23 hrs | 9hrs (10) | 16 hrs ( 4) | 26 mins | 6 mins (20) | 10 mins (20) |

Table 3: Breakdown of indexing latency for transductive $\text{MF}_{\text{TRNS}}$ and inductive $\text{MF}_{\text{IND}}$ matrix factorization methods on SciDocs and Hotpot-QA. For each setting, we show the number of epochs for training the model in parentheses. Total indexing time also includes the time taken to compute initial query and item embeddings using $\text{DE}_{\text{SRC}}$. Computing item embeddings takes 90 seconds for SciDocs (with 25K items) and ~2 hours for Hotpot-QA (with 5 million items) on an Nvidia 2080ti GPU with 12 GB GPU memory.

Figure 10 shows Top-$k$-Recall versus indexing time for and MF with two different strategies to construct sparse matrix $G$ and Table 3 shows the time taken to construct the sparse matrix $G$ and the time taken to train the matrix factorization model. $\mathcal{Q} - *$ indicates that $G$ is constructed by selecting a fixed number of $k_d$ items per *query* in $\mathcal{Q}_{\text{train}}$, and $\mathcal{I} - *$ indicates that $G$ is constructed by selecting fixed number of $k_d$ queries per *item* in $\mathcal{I}$. When selecting a fixed number of items per query, we experiment with $|\mathcal{Q}_{\text{train}}| \in \{1K, 10K, 50K \}$ and $k_d \in \{100, 1000\}$. When selecting a fixed number of queries per item, we first create a pool of 50K queries and then select $k_d$ queries per item for $k_d \in \{2, 5, 10\}$.

**Transductive Matrix Factorization** For $\text{MF}_{\text{TRNS}}$, both $\mathcal{Q}-*$ and $\mathcal{I}-*$ strategies yield similar Top-$k$-Recall at a given indexing cost on SciDocs as both strategies result in each item being scored with at least a few queries. However, on Hotpot-QA, selecting a fixed number of items per query may not result in each item being scored against some queries, and thus $\mathcal{Q} - *$ variants yield marginal (if any) improvement over $\text{DE}_{\text{SRC}}$. $\mathcal{I} - *$ variants perform better than $\text{DE}_{\text{SRC}}$ and corresponding $\mathcal{Q} - *$ variants as each item is scored against a fixed number of queries. Note that this performance improvement comes at the cost of an increase in time required to compute sparse matrix $G$, as shown in Table 3.

**Inductive Matrix Factorization** For $\text{MF}_{\text{IND}}$, we observe that $\mathcal{Q} - *$ variants consistently provide better recall-vs-indexing time trade-offs as compared to corresponding $\mathcal{I} - *$ variants on both SciDocs and Hotpot-QA.

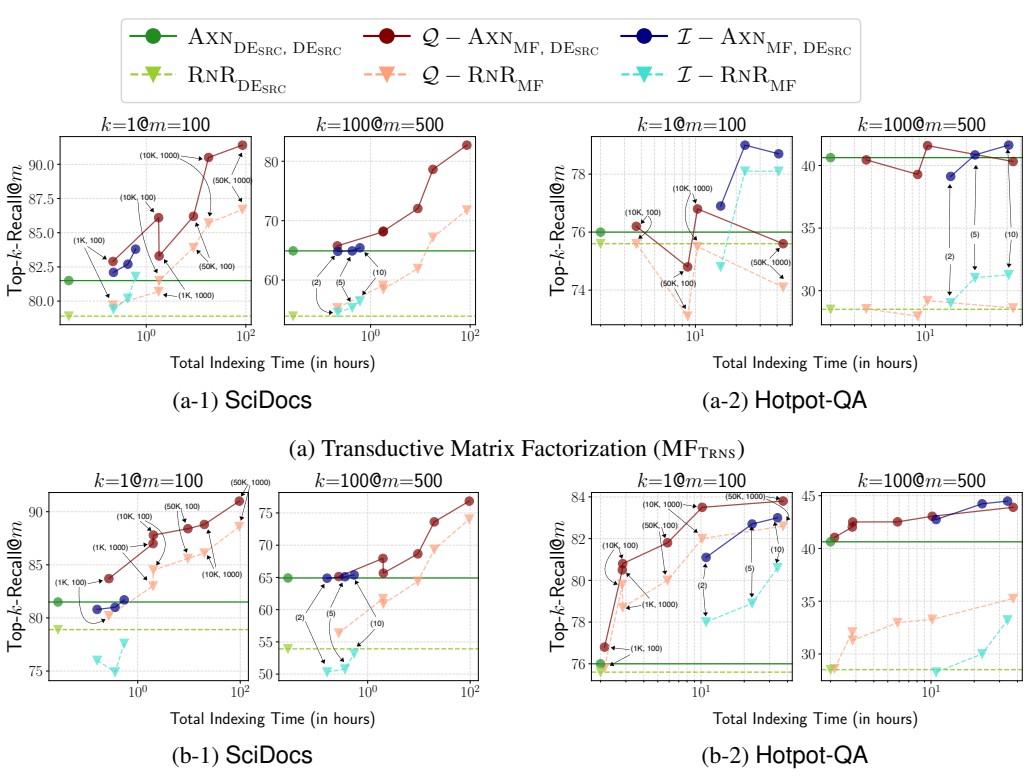

Figure 10: Top-1-Recall and Top-100-Recall at fixed inference cost budget ($m$) of 100 and 500 cross-encoder calls respectively versus indexing time (in hours) for different strategies of constructing sparse matrix $G$. $\mathcal{Q} - *$ indicates that $G$ is constructed by selecting a fixed number of items per *query* in $\mathcal{Q}_{\text{train}}$, and $\mathcal{I} - *$ indicates that $G$ is constructed by selecting fixed number of queries per *item* in $\mathcal{I}$. For $\mathcal{Q} - *$ approaches, the text annotations indicate ($|\mathcal{Q}_{\text{train}}|, k_d$) pairs where $|\mathcal{Q}_{\text{train}}|$ is the number of anchor/train queries and $k_d$ is the number of items per query in the sparse matrix $G$. For $\mathcal{I} - *$ approaches, the text annotations indicate the number of queries per item in the sparse matrix $G$. See §B.4 for detailed discussion.

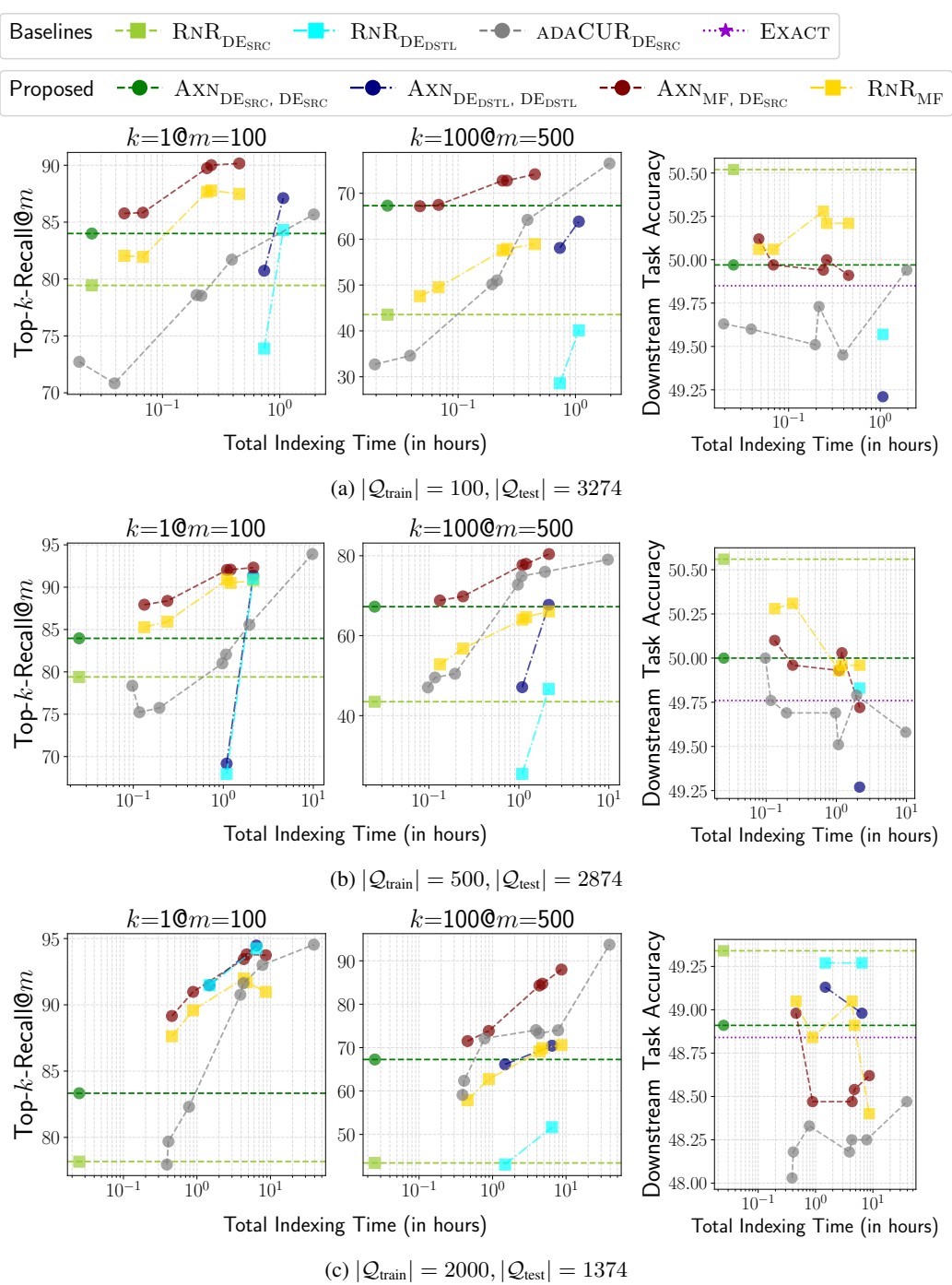

Figure 11: Top-$k$-Recall and downstream task accuracy versus indexing time for various approaches on domain=YuGiOh. We report Top-1-Recall and Top-100-Recall at fixed inference cost budget ($m$) of 100 and 500 CE calls respectively, and downstream task accuracy for fixed inference cost of 100 CE calls. Each subfigure shows results for different train/test splits.

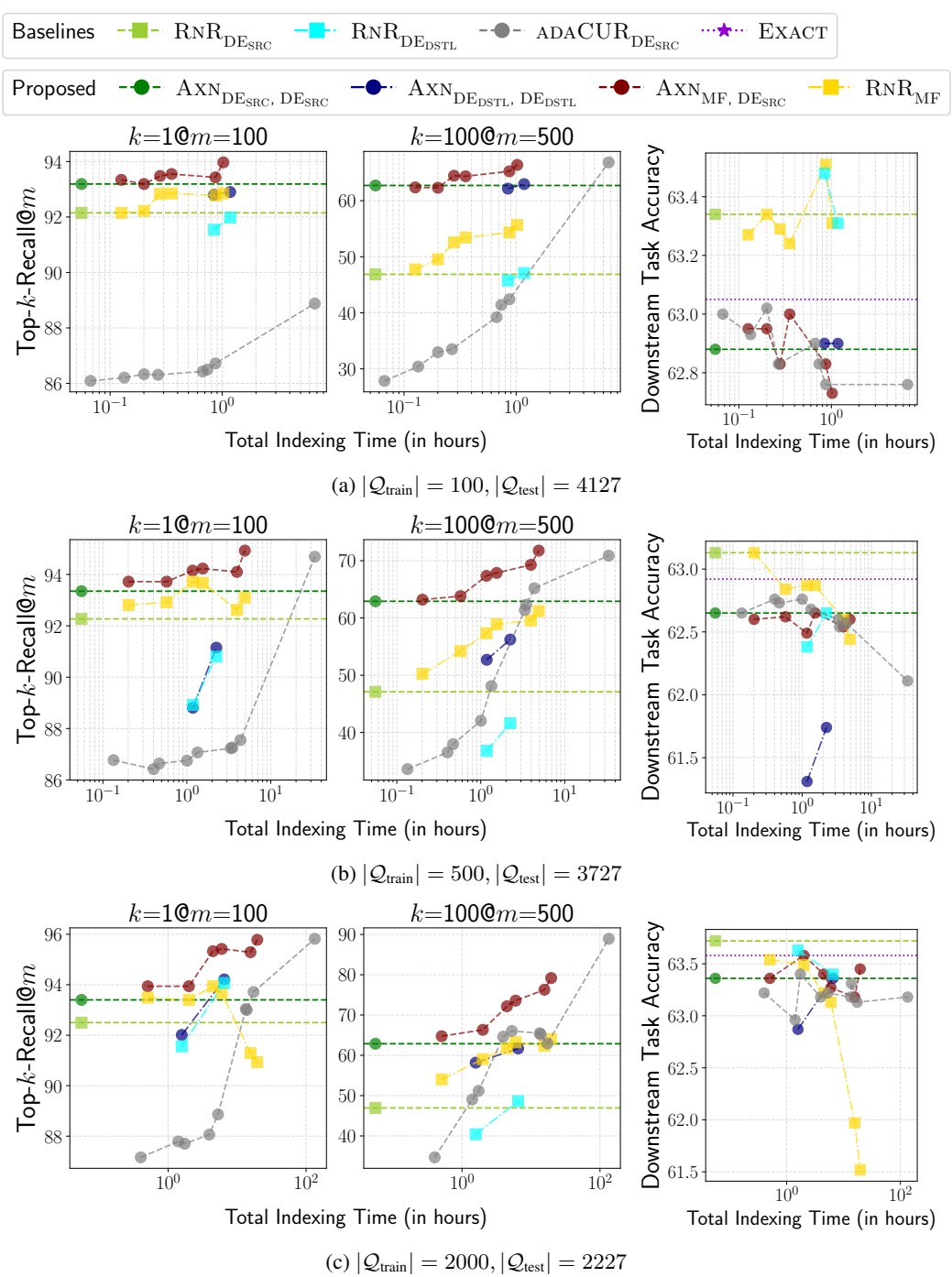

Figure 12: Top-$k$-Recall and downstream task accuracy versus indexing time for various approaches on domain=Star Trek. We report Top-1-Recall and Top-100-Recall at fixed inference cost budget ($m$) of 100 and 500 CE calls respectively, and downstream task accuracy for fixed inference cost of 100 CE calls. Each subfigure shows results for different train/test splits.

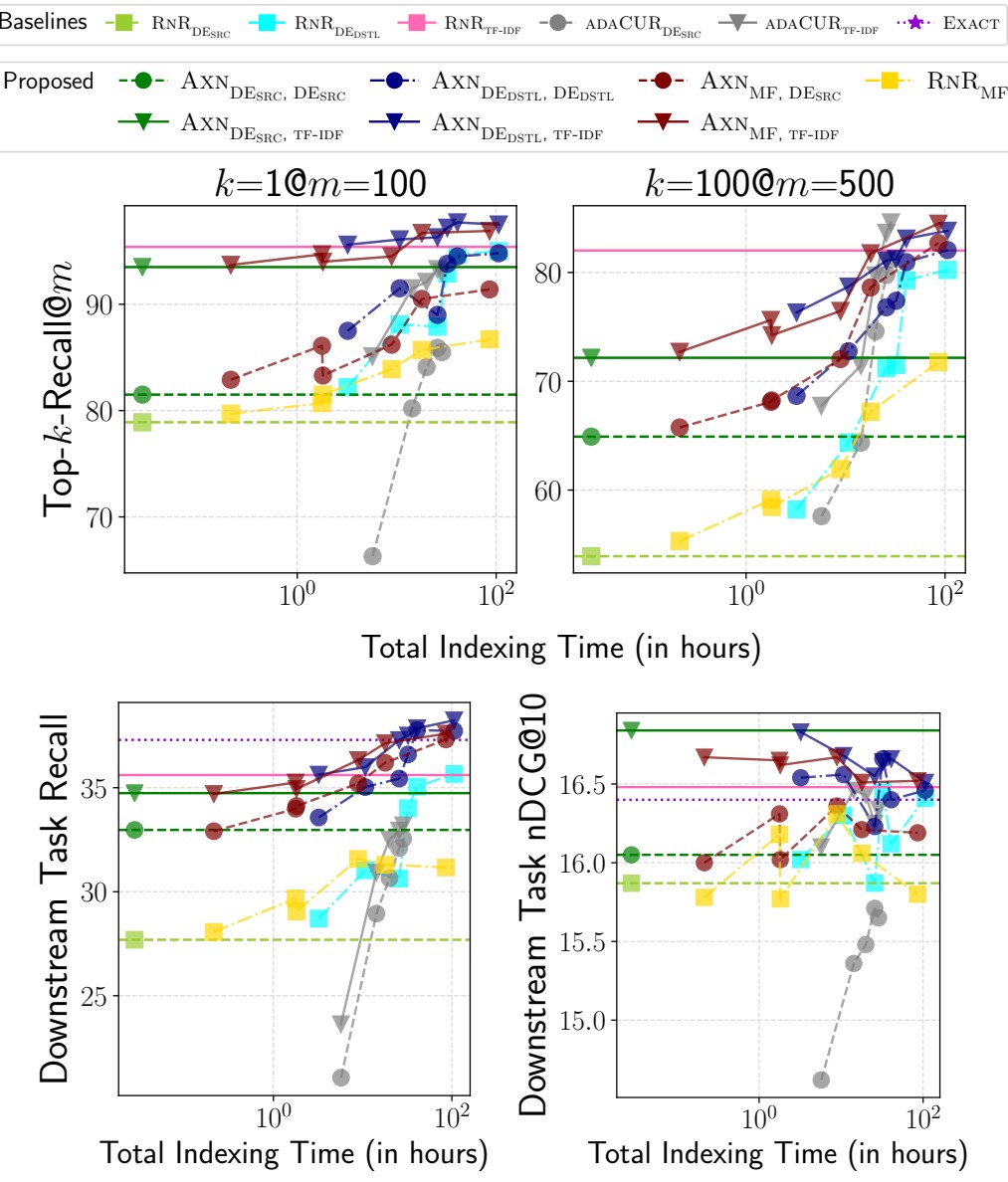

Figure 13: Top-$k$-Recall and downstream task performance metrics versus indexing time for various approaches on domain=SciDocs. We report Top-1-Recall and Top-100-Recall at fixed inference cost budget ($m$) of 100 and 500 cross-encoder (CE) calls respectively, and downstream task metrics for fixed inference cost of 100 cross-encoder calls. We report results for transductive matrix factorization ($MF_{TRNS}$) in these plots. The base dual-encoder ($DE_{SRC}$) in these plots is a 6-layer distilbert model finetuned on MS-MARCO dataset. The $DE_{SRC}$ model is available at https://huggingface.co/sentence-transformers/msmarco-distilbert-base-v2.

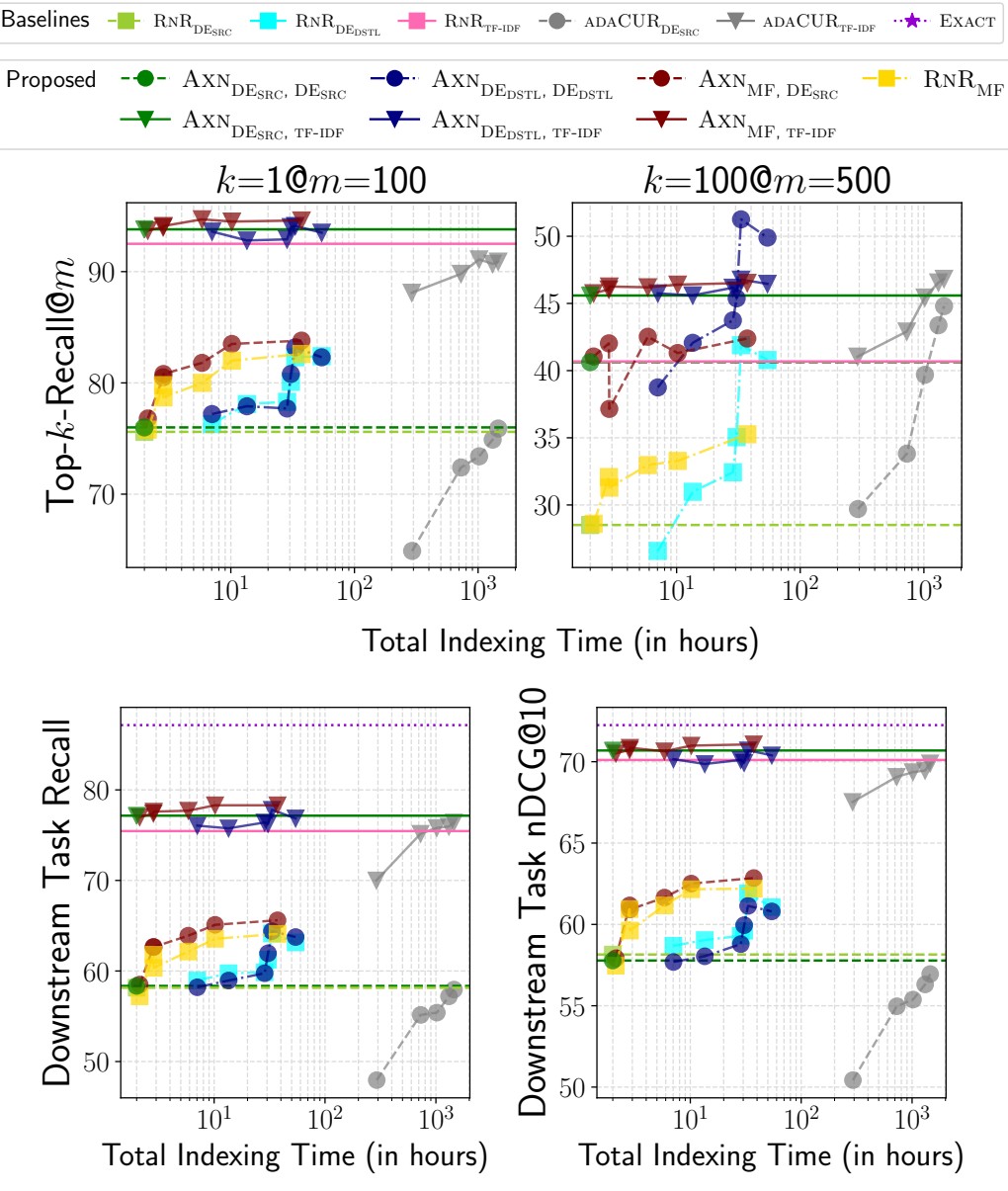

Figure 14: Top-$k$-Recall and downstream task performance metrics versus indexing time for various approaches on domain=Hotpot-QA. We report Top-1-Recall and Top-100-Recall at fixed inference cost budget ($m$) of 100 and 500 cross-encoder (CE) calls respectively, and downstream task metrics for fixed inference cost of 100 cross-encoder calls. We report results for inductive matrix factorization ($\text{MF}_{\text{IND}}$) in these plots. The base dual-encoder ($\text{DE}_{\text{SRC}}$) in these plots is a 6-layer distilbert model finetuned on MS-MARCO dataset. The $\text{DE}_{\text{SRC}}$ model is available at https://huggingface.co/sentence-transformers/msmarco-distilbert-base-v2.

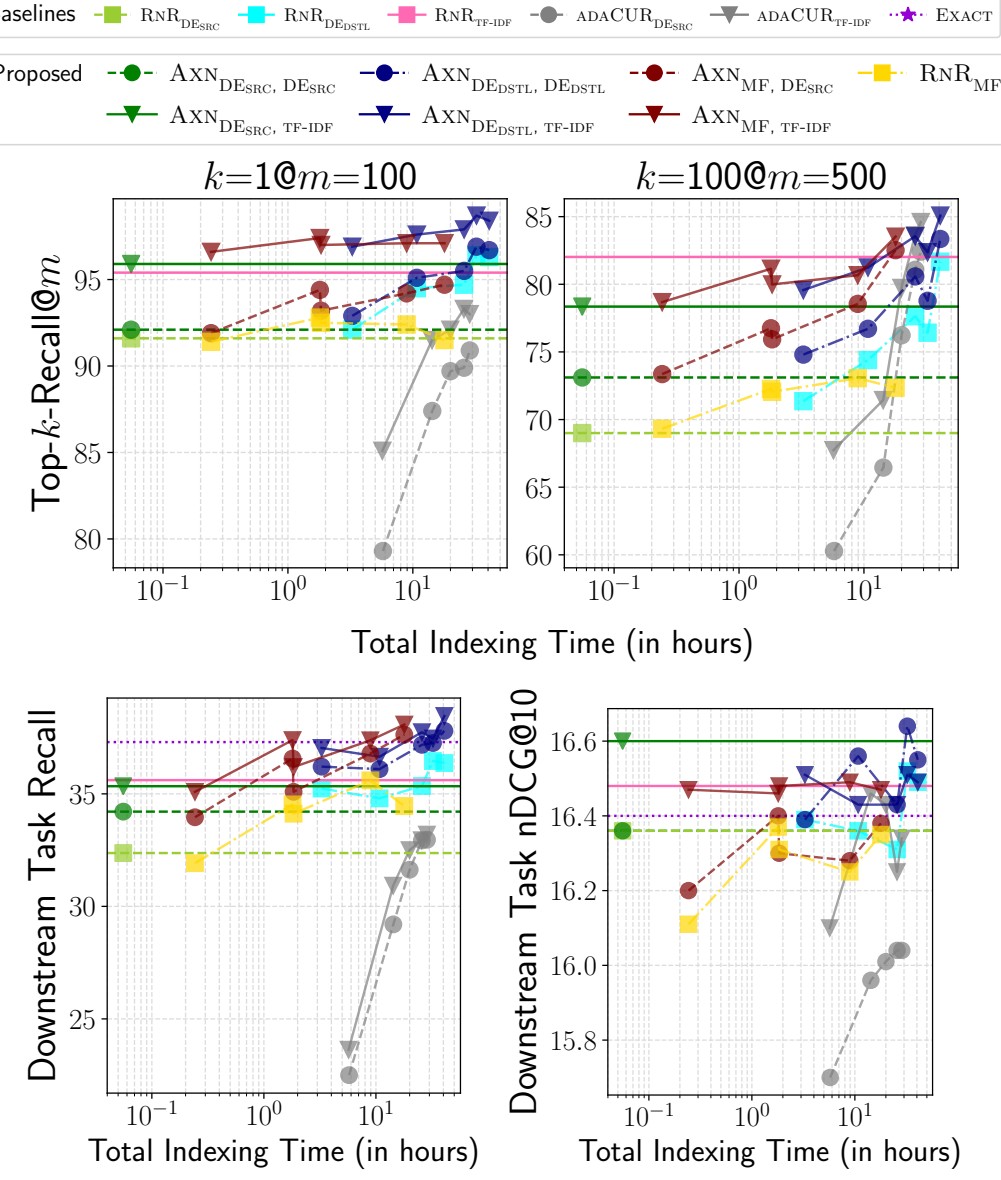

Figure 15: Top-$k$-Recall and downstream task performance metrics versus indexing time for various approaches on domain=SciDocs. We report Top-1-Recall and Top-100-Recall at fixed inference cost budget ($m$) of 100 and 500 cross-encoder (CE) calls respectively, and downstream task metrics for fixed inference cost of 100 cross-encoder calls. We report results for transductive matrix factorization ($MF_{TRNS}$) in these plots. The base dual-encoder ($DE_{SRC}$) in these plots is a 12-layer bert-base model finetuned on MS-MARCO dataset. The model is available at https://huggingface.co/sentence-transformers/msmarco-bert-base-dot-v5.

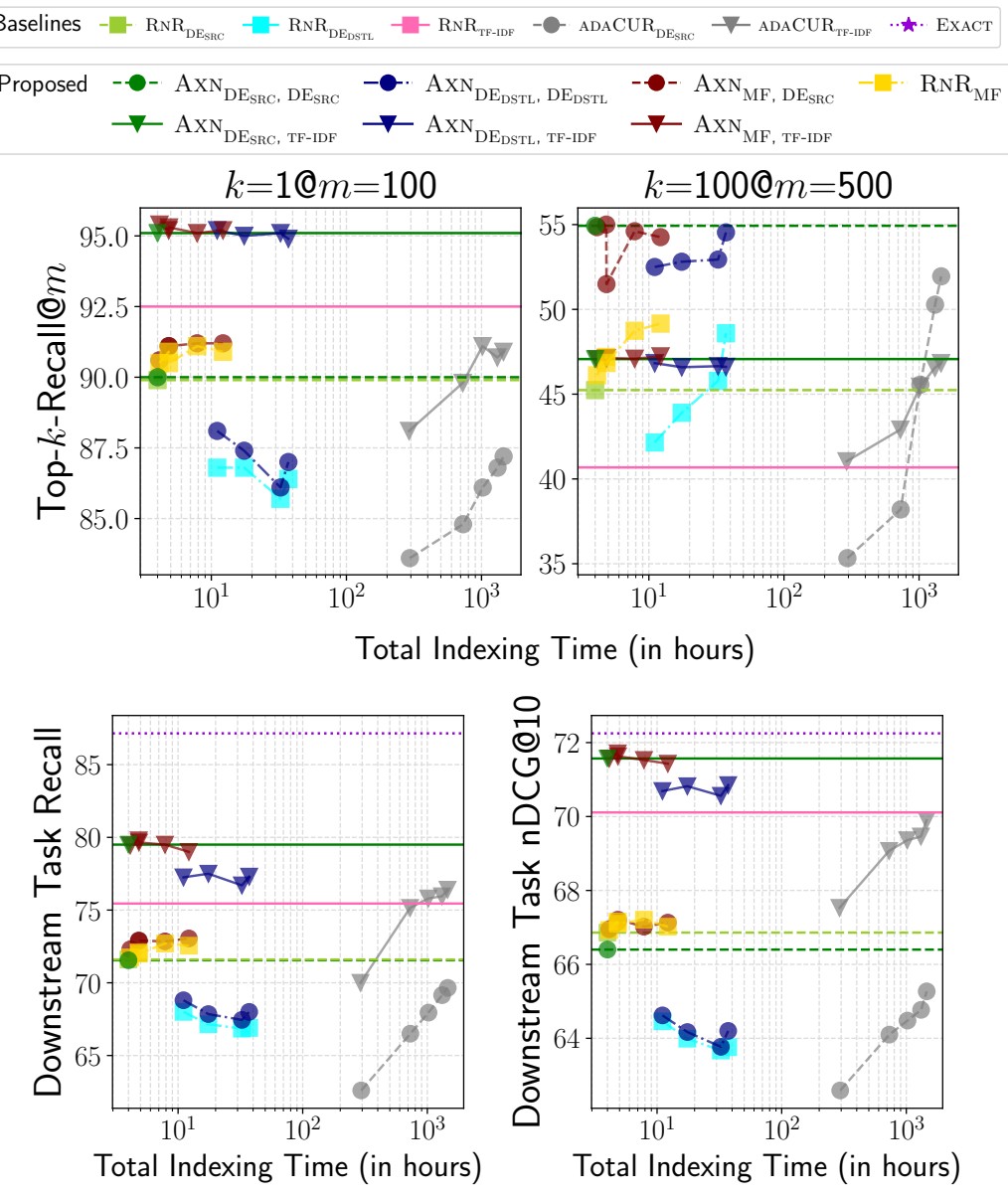

Figure 16: Top-$k$-Recall and downstream task performance metrics versus indexing time for various approaches on domain=Hotpot-QA. We report Top-1-Recall and Top-100-Recall at fixed inference cost budget ($m$) of 100 and 500 cross-encoder (CE) calls respectively, and downstream task metrics for fixed inference cost of 100 cross-encoder calls. We report results for inductive matrix factorization (MF$_{\text{IND}}$) in these plots. The base dual-encoder (DE$_{\text{SRC}}$) in these plots is a 12-layer bert-base model finetuned on MS-MARCO dataset. This DE$_{\text{SRC}}$ model is available at https://huggingface.co/sentence-transformers/msmarco-bert-base-dot-v5.

