# OpenReview forum: "Adaptive Retrieval and Scalable Indexing for k-NN Search with Cross-Encoders"
_ICLR.cc/2024/Conference — ICLR 2024 poster_

### Official Review · Reviewer_hzee · 2023-10-31

**Soundness:** 3 good
**Presentation:** 2 fair
**Contribution:** 2 fair
**Rating:** 5
**Confidence:** 4

**Summary:**

The paper proposes AXN, a test-time multi-run query embedding adaptation approach that leverages KNN search to approximate cross-encoder scores. This method successfully reduces the expensive computational costs associated with cross-encoder calculations, surpassing the performance of DE-based and CUR-based alternatives.

**Strengths:**

The paper presents a novel adaptive retrieval technique that utilizes a limited number of cross-encoder (CE) calls to approximate the quality of cross-encoder results. This approach is scalable to handle a large volume of items.

**Weaknesses:**

1)	This paper uses K nearest neighbor search to iteratively update query embedding and approximate cross-encoder results. However, many references of nearest neighbor search are missing.
2)	AXN utilizes sparse matrix to reduce index costs. The paper lacks detailed analysis regarding this technique's impact on results concerning varying degrees of sparsity.
3)	Both AXN and CUR-based methods need to compute low-dimensional embeddings for queries and items. AXN uses sparse matrix to reduce the cost. This can also be applied to CUR-based methods to reduce the index time. It is unclear if other techniques of AXN generate substantial improvements over CUR-based methods. It would be great if the authors can add more ablation study experiments.
4)	The paper only covers the total index time as a benchmark, and future exploration could include query latency measurements since various steps are executed many times, including Solve-Linear-Regression and topk search.
5)	It introduces lambda to ensemble the generated query embedding with a query embedding from DE or inductive matrix factorization. However, it fails to conduct an analysis of lambda impact on evaluation. It is the same for all experiments, or should be tuned in each experiment?
6)	It runs R times Solve-Linear-Regression and topk search. How to choose R? It is fixed in all experiments or should be tuned in each experiment? Should it be large in large dataset?
7)	The same problem for hyper-parameter Ks.

**Questions:**

1)	“CUR” appears without any definition.
2)	What is the topk search method? Is it brute-force search?
3)	Please address the above weaknesses.

---

> ### Author Response · Authors · 2023-11-15
> **Response to Reviewer hzee (Part 1)**
>
> Thank you for your time and valuable feedback. We hope that our response helps to address the points you’ve raised. We would be happy to further discuss and clarify any remaining questions/concerns.
>
>
> > (1) This paper uses K nearest neighbor search to iteratively update query embedding and approximate cross-encoder results. However, many references of nearest neighbor search are missing
>
>    - In the related work section, we have over 25 citations outlining the relationship between our work and kNN search literature, especially in the **Nearest Neighbor Search** paragraph of related work. This paragraph in related work covers data structures for efficient indexing, speeding up similarity computation (e.g. using quantization), and the combination of graph-based/tree-based indexes and neural models.
>    - Our work focuses on approximating an expensive black-box similarity function (eg. cross-encoders) using matrix factorization approaches to perform kNN search, and we discuss related work in the **Approximating Similarity Function** paragraph in the Related Work section. If you have further suggestions for citations for this paragraph, we greatly appreciate them.
>    - Compared to work on index structures for efficient kNN search (e.g.,CoverTree, HNSW, etc), we view our work as orthogonal (e.g., the vectors produced by our method could be indexed by any such vector-based kNN index). See second paragraph Sec 2.1 of paper for more discussion.
>    - If, given the above comments, the reviewer feels we have inadequately placed our work in the landscape of kNN search, please do let us know. Clearly presenting our work amidst that landscape is important to us. We welcome any further content or citation suggestions that the reviewer might have. Thank you for raising this point to us.
>
>
> > (2) AXN utilizes sparse matrix to reduce index costs. The paper lacks detailed analysis regarding this technique's impact on results concerning varying degrees of sparsity.
>
>    - The ablation you reference is indeed already contained in our paper; it is present both in Table 3 and in Appendix B.4. These experiments vary the number of queries in the sparse matrix as well as number of items scored per query (i.e. sparsity) and measure the change in test-time Top-k-Recall and indexing latency.
>    - In Appendix B.4, we observe that increasing the number of items per query in the matrix (i.e. reducing sparsity), and increasing number of train queries generally leads to improvement in test-time Top-k-Recall although at increased indexing cost, as shown in Figure 10 (in updated pdf). This is an expected consequence; e.g., more observed data improves recall performance.
>    -  In Table 3, we show a detailed breakdown of indexing cost for different configurations of the sparse matrix. Here are some key observations:
>       -  Computing the cross-encoder (CE) scores for query-item pairs in the sparse matrix takes a significant fraction of overall indexing time.
>       -  The time taken to factorize the sparse matrix is proportional to the number of query-item pairs observed in the sparse matrix. However, it remains a small fraction of the overall indexing cost.
>
> > (3) …  AXN and CUR-based methods need to compute low-dimensional embeddings for queries and items. AXN uses sparse matrix to reduce the cost. This can also be applied to CUR-based methods to reduce the index time. …
>
>    - The most significant component of indexing cost is computation of cross-encoder scores between train/anchor queries and items.
>    - To the best of our understanding, the item indexing step for CUR-based methods _must_ compute a _dense matrix_ by scoring each item against _all_ of the anchor queries while we propose use of sparse matrices to significantly bring down the number of query-item scores computed using the cross-encoder.
>    - The indexing cost for AdaCUR can be controlled by the varying number of anchor/train queries, and in Fig 1, we report AdaCUR results for varying number of anchor/train queries.
>    - As shown in Fig 1, at a given indexing cost, our sparse matrix factorization based approaches yield better test-time Top-k-Recall values than AdaCUR.
>    -  Were there other ablations that the reviewer had in mind that might help improve our understanding of where our proposed sparse matrix based approaches outperforms CUR-based methods such as AdaCUR?

---

> > ### Author Response · Authors · 2023-11-15
> > **Response to Reviewer hzee (Part 2)**
> >
> > > (4) The paper only covers the total index time as a benchmark….
> >    - The recall vs test-time inference cost suggested by the reviewer are indeed already present in  Figure 6 (Fig 7 in updated pdf) where AXN consistently outperforms DE-based retrieve-and-rerank baseline for all cost budgets.
> >    - We use the number of cross-encoder (CE) calls made at test-time as the inference cost in Fig 6 (Fig 7 in updated pdf) because we observe that inference latency for both AXN and DE-based retrieve-and-rerank approaches is similar at a given CE call budget.
> >        - This is because computing CE scores for retrieved items at test-time takes most of the time and the overhead for our proposed approach is rather minimal when using 5-10 rounds, as shown in Figure 5.
> >
> >
> > > (5)  …lambda to ensemble the generated query embedding with a query embedding from DE or inductive matrix factorization….
> >    - For ZeShEL, we used $\lambda = 0$ (i.e. ignoring query embedding from DE or inductive MF model),
> >    - For BeIR datasets, we use $\lambda=0.8$ for k=1 and $\lambda=0$ for k=100.
> >      - We picked these values using the dev set for using AXN’s adaptive retrieval with the baseline DE model (see Fig. 8 in updated pdf) and used the same value of
> >    $\lambda$ for AXN in all our experiments such as when used with other matrix factorization methods.
> >    - We refer to Appendix B.2 for ablation on effect of $\lambda$, and to Appendix A.4 for default values of $\lambda$.
> >
> > > (6) … It runs R times Solve-Linear-Regression and topk search. How to choose R? It is fixed in all experiments or should be tuned in each experiment? Should it be large in large dataset? …
> >    - As mentioned in Appendix A.4, we use the number of rounds, R = 10 for BeIR datasets and R = 5 for ZeShEL datasets, unless specified otherwise.
> >    - Figure 5 shows the effect of R on inference latency as well as it shows breakdown of inference latency into various components, and Figure 6 (in the updated pdf) shows the effect of R on Top-k-Recall.
> >    - As shown in Figure 6 (in updated pdf), Top-k-Recall saturates around 5-10 rounds and AXN incurs a very small overhead for 5-10 rounds, as shown in Figure 5.
> >
> > > (7) The same problem for hyper-parameter Ks.
> >
> > We split the CE call budget ($B_{CE}$) uniformly over R rounds i.e. $k_s =  B_{CE}/R$ .
> >
> >
> >
> >
> > ### Response to Questions
> >
> > >“CUR” appears without any definition
> >
> > CUR-matrix factorization approximates a matrix M using a subset of its columns and a subset of its rows. We refer the reviewer to Mahoney & Drineas (2009), Yadav et al. (2022) for a more detailed description.
> >
> > > What is the topk search method? Is it brute-force search?
> >
> > In our current experiments, we use brute-force search for the `topk` search step in Line 8, Algo. 1.
> >    - The “Matrix Multiply” bar in Fig 5 shows the fraction of time spent on `topk` search step.
> >    - Even with brute-force computation over 5 million items, this `topk` search step makes negligible contribution to overall inference latency as we compute this on a GPU and most of the inference time is actually spent in computing cross-encoder scores of retrieved items.
> >    - We believe that it is straightforward to use an off-the-shelf approximate vector-based kNN search method to retrieve top-k based on query and item embeddings if the brute-force computation of Line 8, Algo. 1 becomes a bottleneck, for eg on CPUs or when scaling to billions of items.
> >
> >
> >
> > ### References
> > - [1] Mahoney, Michael W., and Petros Drineas. "CUR matrix decompositions for improved data analysis." Proceedings of the National Academy of Sciences 106.3 (2009): 697-702.
> > - [2] Yadav, Nishant, et al. "Efficient nearest neighbor search for cross-encoder models using matrix factorization." EMNLP 2022.

---

> > > ### Author Response · Authors · 2023-11-21
> > >
> > > Thank you for your time and valuable feedback.
> > >
> > > We have attempted to address the concerns raised by you in our response above. We would really appreciate an appropriate increase in the score to facilitate acceptance of the paper if your concerns have been adequately addressed and we would be happy to further discuss and clarify any remaining questions/concerns.

---

> > > > ### Comment · Reviewer_hzee · 2023-11-23
> > > >
> > > > I appreciate the author's response as it addresses some of my original concerns, and I have consequently increased my score. However, I would like to seek clarification on the following points: 1) Can the author please provide the matrix construction time, matrix factorization time, inference time, and recall comparison between AXN and all baselines to help me understand the detailed advantages over different baselines? 2) It appears that lambda and R should be tuned differently for each dataset and different Ks. Can the author provide insights on how to choose these parameters for a new dataset?

---

> > > > > ### Author Response · Authors · 2023-11-23
> > > > > **Response to follow-up questions by reviewer hzee**
> > > > >
> > > > > We are glad that we were able to clarify your concerns! Here is our response for the follow-up questions,
> > > > > and we will be happy to provide further clarifications if there are any follow-up questions/concerns.
> > > > >
> > > > > > (1) Can the author please provide the matrix construction time, matrix factorization time, inference time, and recall comparison between AXN and all baselines to help me understand the detailed advantages over different baselines?
> > > > >
> > > > >    - We refer the reviewer to Table 3 in the Appendix for detailed numbers on matrix construction
> > > > > time and matrix factorization time for different dimensions and sparsity of the matrix on
> > > > > small-scale and large-scale datasets.
> > > > >       - Overall, matrix construction time was typically the dominant component of the
> > > > >       offline indexing step as it involved computing cross-encoder scores for
> > > > >       observed query-item pairs in the matrix.
> > > > >       - For a given sparse matrix, proposed matrix factorization approaches
> > > > >       can be significantly more efficient that training dual-encoder models
> > > > >       via distillation, as shown in Figure 2.
> > > > >   - Inference Time vs Recall
> > > > >      - We present a direct comparison between AXN (proposed) and DE-based retrieve-and-rerank at varying
> > > > >     inference cost budgets in Figure 6 (Fig 7 in updated pdf) where AXN consistently
> > > > >     outperforms DE-based retrieve-and-rerank baseline.
> > > > >        - We measure inference cost by the number of cross-encoder (CE) calls made at
> > > > >      test-time in Figure 6 (Fig 7 in updated pdf) as inference latency for AXN
> > > > >      (with 5 to 10 rounds) and DE-based retrieve-and-rerank approaches is similar
> > > > >      at a given CE call budget.
> > > > >        - The reason behind similar inference latency for AXN (with 5 to 10 rounds)
> > > > >      and DE-based retrieve-and-rerank is that the inference latency overhead
> > > > >      incurred by AXN is less than 2 to 5% of overall time when using 5-10 rounds
> > > > >      for inference and the majority of time is spent in computing cross-encoder
> > > > >      scores for the retrieved items, as shown in Figure 5.
> > > > >      - In other figures (such as Figure 1,3, 11-16), we present comparison at a
> > > > >      _fixed_ inference cost while varying indexing cost on x-axis to highlight the
> > > > >      indexing efficiency of proposed approaches over baselines.
> > > > >        - We observe that proposed approaches can be up to 100x and 5x more efficient
> > > > >        than AdaCUR and DE-distillation approaches respectively in terms of indexing cost
> > > > >        while improving or matching kNN recall values.
> > > > >
> > > > >
> > > > >
> > > > > > (2) It appears that lambda and R should be tuned differently for each dataset and different Ks.
> > > > > Can the author provide insights on how to choose these parameters for a new dataset?
> > > > >
> > > > > > Choosing $R$
> > > > >   - We picked $R$ values based on empirical observation that recall of proposed approach (AXN)
> > > > > saturates around 5-10 rounds (as shown in Fig 6) while incurring less than 5% overhead (as shown in Fig 5).
> > > > >   - We used same value of $R$ of all inference cost budgets.
> > > > >   - For BeIR datasets (hotpotqa and scidocs), we used $R$ = 10 and
> > > > >   for entity linking datasets (yugioh and star_trek), we used $R$=5.
> > > > >
> > > > >  - We observed that increasing $R$ to large values such as 50 or 100 yield negligible
> > > > > improvement in recall (see Fig 6) while significantly increasing the overhead by up to 25-40% (as shown in Fig 5).
> > > > >   - For a new dataset, we recommend using value between 5 to 10 based on above observations.
> > > > >
> > > > > > Choosing $\lambda$
> > > > >  - We picked value of $\lambda$ parameter by tuning on dev set on domain=HotPotQA
> > > > >   as shown in Figure 8 and we used the same values for domain=Scidocs.
> > > > >   - For entity linking dataset (yugioh and star-trek), we used $\lambda=0$.

---

### Official Review · Reviewer_ujET · 2023-11-01

**Soundness:** 3 good
**Presentation:** 4 excellent
**Contribution:** 3 good
**Rating:** 6
**Confidence:** 3

**Summary:**

Cross-encoder (CE) models outperform Dual-encoder (DE) models (especially at zero-shot problems) in the ranking task but are very expensive to use during inference. To alleviate this usually a retrieve then re-rank approach is used where a set of items are first retrieved using a DE model and then further ranked by CE. This paper proposes an alternate approach where the CE model is first distilled into a lightweight factorized model and at test time query representation is iteratively fine-tuned such that the dot product between test query embedding and indexed item embeddings gets closer and closer to the CE assigned relevance. This approach helps in reducing the CE calls required to accurately rank items for the test query.

**Strengths:**

- The approach is simple to plug into existing retrieval and ranking frameworks
- The paper is in general well-written and easy to follow
- The proposed approach is compared against relevant baselines and the evaluation is thorough

**Weaknesses:**

- The proposed approach is evaluated only on zero-shot tasks, does this approach also benefits standard retrieval tasks
- Gains are primarily under fixed index time scenario which is usually a one-time cost

**Questions:**

- It is a bit surprising that RnR DE models are performing worse than TF-IDF on Hotpot, is it because this CE model was trained on triplets mined using TF-IDF?
- CE model is trained conditioned on some specific negativing mining distribution so maybe for RnR baselines we should also compare with a retrieval model which is the same as the retrieval model used for the negative mining so that the train and test-time behaviours are same
- For a given budget $X$ of CE calls, how should one distribute $X$ calls in the number of rounds and $K$ CE calls inside each round in AXN inference

---

> ### Author Response · Authors · 2023-11-15
> **Response to Reviewer ujET**
>
> Thank you for your time and valuable feedback. We have attempted to address the concerns raised by the reviewer and we would be happy to further discuss and clarify any remaining questions/concerns.
>
> > Gains are primarily under fixed index time scenario which is usually a one-time cost
>
> - While indexing is a one-time cost, it is a significant one. In particular, it is crucial to scaling to large numbers of items in terms of time and compute resources.
>    - Existing approaches such as  CUR-based methods do not scale well to millions of items and can require 1000s of GPU hours,
> and training parametric dual-encoder models via distillation also requires significant amount of time and multiple GPUs with
> large amounts of memory.
>    - In contrast, our proposed matrix factorization based approaches incur up to 100x and 5x less indexing time as compared to AdaCUR or dual-encoder distillation-based approaches respectively.
>
> - In addition to proposing time- and resource-efficient indexing methods, we propose an adaptive retrieval method (AXN)
> that yields significant improvement over existing dual-encoder (DE) based retrieve-and-rerank pipelines,
> - Moreover, AXN can be used in combination with _any_ existing DE model to improve test-time performance _without incurring any
> additional indexing cost_.
>
> > The proposed approach is evaluated only on zero-shot tasks, does this approach also benefits standard retrieval tasks
>
> - We focus on zero-shot tasks as poor performance of dense-embedding models in zero-shot setting motivates the
> use expensive cross-encoder models.
> - As shown in Table 2 in Thakur et al. (2021), there is a substantial gap of about 15 points between zero-shot performance of dual-encoder and cross-encoder models.
> - As reported in Table 2 in Thakur et al. (2021), using cross-encoder models yield marginal (about 1%) improvements over dual-encoder/dense
> embedding models on _in-domain_ datasets such as MS-MARCO, and the marginal gains
> may not justify using cross-encoders for retrieval in such _in-domain_ settings.
>
>
> ### Response to Questions
> > It is a bit surprising that RnR DE models are performing worse than TF-IDF on Hotpot, is it because this CE model was trained on triplets mined using TF-IDF?
>    - As shown and reported in prior work (Thakur et al., 2021), TF-IDF is a strong baseline for these _zero-shot_ information retrieval tasks
>    and dual-encoders typically perform worse than TF-IDF by 3-4% on average (see Table 2 in Thakur et al. (2021)).
>    - While TF-IDF performing better than dual-encoders on downstream task metric is a result of the nature of the task and data, we  hypothesize that TF-IDF also performing better at kNN search with cross-encoder indicates that the cross-encoder is aligned with the TF-IDF similarity measure to some extent.
>        - We hypothesize that this alignment could be a result of  both the training strategy used for training the cross-encoder model as well as the nature of the task and data.
>
> > CE model is trained conditioned on some specific negativing mining distribution so maybe for RnR baselines we should also compare with a retrieval model which is the same as the retrieval model used for the negative mining so that the train and test-time behaviours are same
>
>    - We agree that the training strategy used for training the cross-encoder can play a critical role in the performance of various baselines.
>    - A detailed study of how the cross-encoder training strategy affects performance of various retrieval methods is an interesting research question but beyond the scope of this paper.
>    - For ZeShEL datasets, the CE model is trained using negatives retrieved using a separately trained dual-encoder (DE), and this is the same DE model that is used in the DE-based retrieve-and-rerank baseline reported in the paper. We used the CE model released by Yadav et al. (2022) in our experiments.
>    - For BeIR,  the baseline DE model is trained on the same set of labeled triplets as the cross-encoder model.
>
>
> > For a given budget of X CE calls, how should one distribute X calls in the number of rounds and K CE calls inside each round in AXN inference
>
> **Distributing CE call budget over multiple rounds**
>    - In our experiments, we distributed the CE call budget uniformly over each round.
>    - We observed that for a given CE call budget, increasing the number of rounds leads to improvements in Top-k-Recall
>    metrics (as reported in Fig 6 in updated pdf) at the expense of increased overhead (as reported in Figure 5)
>    with the Top-k-Recall saturating around 5-10 rounds.
>    - We use 10 rounds for BeIR domains and 5 rounds for ZeShEL domains.
>
> ### References:
> - Thakur, Nandan, et al. "BeIR: A heterogenous benchmark for zero-shot evaluation of information retrieval models." NeurIPS 2021 Dataset and Benchmark Track.
> - Yadav, Nishant, et al. "Efficient nearest neighbor search for cross-encoder models using matrix factorization." EMNLP 2022.

---

> > ### Comment · Reviewer_ujET · 2023-11-23
> >
> > Thanks for the response, it added to my understanding of the paper, I would like to maintain my original score.

---

### Official Review · Reviewer_ja98 · 2023-11-01

**Soundness:** 4 excellent
**Presentation:** 3 good
**Contribution:** 3 good
**Rating:** 8
**Confidence:** 3

**Summary:**

The authors proposed a sparse-matrix factorization-based approach to improve the efficiency of fitting an embedding space to approximate the cross-encoder for k-NN search. Unlike DE-based and CUR-based methods, which lack good generalizations and computation efficiency, the new AXN method constructs a sparse matrix containing a cross-encoder score of training queries and all items.  The item embeddings are learned from matrix factorization. During test time, AXN alternates between updating the query embedding and retrieving more items for k-NN indexing.

**Strengths:**

1. The authors proposed AXN, a novel cross-encoder-based k-NN search algorithm. By learning item embeddings from sparse matrix factorization and fixing them during query time, the algorithm is more computationally efficient than other methods.
2. The authors explained their method very clearly in section 2.
3. The extensive experiments and ablation studies supported their claims.

**Weaknesses:**

1. Figure 1's legends and corresponding subplots are hard to read. The subplots are too small, and hard to map points to the legends.

**Questions:**

1. In the experiment section it was not clear how many rounds of updates are performed in all AXN experiments.
2. Consider fixing max of CE calls B_{CE}, but varying the number of iterative search rounds and the number of items to retrieve in each round. Will it affect AXN's performance and total indexing time?

---

> ### Author Response · Authors · 2023-11-15
> **Response to Reviewer ja98**
>
> Thank you for your time, thoughtful review, and support of our paper.
>
> > Figure 1's legends and corresponding subplots are hard to read. The subplots are too small, and hard to map points to the legends.
>
> We have included a larger version of the figures in the Appendix (Figures 9 through 12 in the original submission and Figures 11 through 16 in the updated pdf) as we were able to enlarge the figure 1 in the main paper marginally given the current page limit constraints.
>
> ### Response to Questions:
> >In the experiment section it was not clear how many rounds of updates are performed in all AXN experiments.
>  - As mentioned in Appendix A.4, we use 10 rounds for BeIR datasets and 5 rounds for ZeShEL datasets, unless specified otherwise.
>
> > Consider fixing max of CE calls B_{CE}, but varying the number of iterative search rounds and the number of items to retrieve in each round. Will it affect AXN's performance and total indexing time?
>    - We added additional results in Figure 6 which show that, for a fixed CE call budget, performance of AXN improves as we increase the number of iterative search rounds, with the performance saturating around 5-10 rounds. We divide the CE call budget uniformly over each round.
>   - Figure 5 shows the trend for how the overhead for AXN varies under different test-time cost budgets ( $B_{CE}$ ), and we observe that AXN incur 2 to 5% overhead when using 5-10 rounds for inference.
>   -  The offline indexing time is independent of maximum test-time CE call budget, and depends on the number of query-item pairs scored using the cross-encoder in the sparse matrix as well as time taken to factorize the matrix during the offline indexing stage.

---

> > ### Comment · Reviewer_ja98 · 2023-12-01
> >
> > Thanks for the response. I will maintain my original score.

---

### Official Review · Reviewer_fa41 · 2023-11-01

**Soundness:** 3 good
**Presentation:** 2 fair
**Contribution:** 3 good
**Rating:** 6
**Confidence:** 3

**Summary:**

This paper considers a new approach to retrieval and indexing that attempts to match the accuracy of cross encoders with lesser training time. Cross-encoders based retrieval allows for computing a relevance function over query and each point in the retrieval corpus and finding the point(s) most relevant. As this are expensive for inference, in practice, dual encoders with separate encoding stacks for query and corpus are used with k-NN used to quickly find the most relevant documents. A compromise on cross-encoder is a CUR decomposition of the relevance signal. This paper proposed a method (AXN) that improves upon recall of dual encoder methods and is much faster than CUR based approaches.

AXN starts with a dual encoder (treated as black box) and iteratively mines for items near a query using the query's representation. Then it uses a cross encoder to score the limited set of items retrieved. It uses this to refine the query representation, search for another limited set of items and so on. The method limits the cross attention to far fewer items than the corpus size. Experiments show that AXN improves the recall of dual encoder and can match the recall of CUR methods.

**Strengths:**

1. AXN yields better recall than dual encoders with faster training times compared to CUR methods
2. They can build on any dual encoder methods (although experiments don't say if they improve on DE encoders)

**Weaknesses:**

1. The inference time and its comparison to simple DE+kNN approaches is not made clear. Would a large dual encoder with same inference time as AXN match its recall? Some of this is tucked away in the appendix in Fig 5 and 6 which seems to suggest the margins between DE and AXN are low normalized for inference cost.
2. The choice of DE is not discussed. Is it completely irrelevant? Is it being compared to state-of-the-art encoders?

**Questions:**

1. would a large dual encoder with same inference time as AXN match it's recall? Or would DE saturate well before AXN.
2. What is the unit on x-axis in Figure 6?
3. Could you talk about the scaling properties of your algorithm as the corpus grows larger?
4. Why are not all methods not represented on each sub-figure in Figure 3?

---

> ### Author Response · Authors · 2023-11-15
> **Response to Reviewer fa41 (Part 1)**
>
> Thank you for your time and valuable feedback. We have attempted to address the concerns raised by the reviewer and we would be happy to further discuss and clarify any remaining questions/concerns.
>
> ### Inference Time for AXN vs DE-based retrieve-and-rerank
>
> > The inference time and its comparison to simple DE+kNN approaches is not made clear. Would a large dual encoder with the same inference time as AXN match its recall? Some of this is tucked away in the appendix in Fig 5 and 6 which seems to suggest the margins between DE and AXN are low normalized for inference cost.
>
> - We present a direct comparison between AXN and DE-based retrieve-and-rerank at varying inference cost budgets in Figure 6 (Fig 7 in updated pdf) where AXN consistently outperforms DE-based retrieve-and-rerank baseline.
> - We measure inference cost by the number of cross-encoder (CE) calls made at test-time in Figure 6 (Fig 7 in updated pdf) as inference latency for AXN (with 5 to 10 rounds) and DE-based retrieve-and-rerank approaches is similar at a given CE call budget.
>     -   The reason behind similar inference latency for AXN (with 5 to 10 rounds) and DE-based retrieve-and-rerank is that the inference latency overhead incurred by AXN is less than 2 to 5% of overall time when using 5-10 rounds for inference and the majority of time is spent in computing cross-encoder scores for the retrieved items, as shown in Figure 5.
> - Using a larger dual-encoder would not increase the inference time significantly as the main test-time bottleneck is computing CE scores for retrieved items.
>     - We add additional results for a larger dual-encoder (Fig 27 and 28 in updated pdf) where we observe AXN continues to outperform DE-based retrieve-and-rerank baseline. See second bullet in **Choice of Dual-Encoder** paragraph for details.
>
>
> ### Choice of Dual-Encoder
>
> > The choice of DE is not discussed. Is it completely irrelevant? Is it being compared to state-of-the-art encoders?
> > would a large dual encoder with same inference time as AXN match it's recall? Or would DE saturate well before AXN.
>
> We run experiments with different dual-encoder baselines and justify the choices below
> - For ZeShEL dataset, the dual-encoder model used is representative of the state-of-the-art model.
> - For BeIR benchmark, we used a [dual-encoder model](https://huggingface.co/sentence-transformers/msmarco-distilbert-base-v2) released publicly that is representative of the state-of-the-art model under the setting which _does not_ involve any distillation-based training on MS-MARCO dataset.
>   - We refer the reviewer to Appendix A.2.1 for more details of the dual-encoder models used in this paper.
>
> - We report additional results in Figures 27 and 28 (in updated pdf) for another larger [dual-encoder model](https://huggingface.co/sentence-transformers/msmarco-bert-base-dot-v5) which is representative of the state-of-the-art on MS-MARCO
> dataset (see [here](https://www.sbert.net/docs/pretrained_models.html#msmarco-passage-models)).
>   - This is bert-base model finetuned on MS-MARCO using ground-truth data as well as further trained through several rounds of distillation using a cross-encoder (different from the ones used in this paper).
>   - We observe similar trends such as
>     - Adaptive retrieval (AXN) outperforms retrieve-and-rerank style inference, and
>     - Sparse matrix factorization based approaches can offer improved performance as compared to the dual-encoder.
>
> - Note that we additionally also compare with $DE_{Dstl}$, dual-encoder models further finetuned on the _target_ domain via distillation using the _given cross-encoder model_.
>   - We find that such distillation based training is compute- and resource-intensive.
>   - In contrast, our proposed sparse matrix factorization based approaches can be up to 5x faster while matching or improving over the performance of distilled dual-encoder models.
>   - Also, as shown in Figure 1, using AXN with $DE_{Dstl}$, outperforms retrieve-and-rerank approach (RnR) with the same distilled dual-encoder  $DE_{Dstl}$ for k=100 and matches performance for k=1.

---

> > ### Author Response · Authors · 2023-11-15
> > **Response to Reviewer fa41 (Part 2)**
> >
> > ### Response to Questions
> > > would a large dual encoder with same inference time as AXN match it's recall? Or would DE saturate well before AXN.
> >
> >   See `Choice of Dual-Encoder` Part 1 response above .
> >
> > > What is the unit on x-axis in Figure 6?
> >
> >  Inference cost in Figure 6 (now Fig 7 in updated pdf) refers to the number of cross-encoder (CE) calls made at test-time.
> >    - Dual-Encoder-based retrieve-and-rerank baselines use the entire CE call budget for re-ranking retrieved items, and
> >    - AXN uses the budget for adaptive retrieval over multiple rounds.
> >
> > > Could you talk about the scaling properties of your algorithm as the corpus grows larger?
> >    - **Indexing time wrt corpus size**:
> >      - The indexing time and test-time performance of our proposed matrix factorization scales well with the corpus size.
> >      - As shown in Appendix B.3,  transductive matrix factorization works better for smaller domains (eg. scidocs) and inductive matrix factorization approaches work better for large domains (eg. hotpotqa with 5 million items).
> >      - Overall, our proposed matrix factorization based approaches incur up to 100x and 5x less indexing time as compared to AdaCUR or dual-encoder distillation-based approaches respectively.
> >      - See Figure 2 for detailed breakdown and comparison of indexing time for proposed matrix factorization approaches and dual-encoder distillation baselines. Some observations:
> >         -  Computing the cross-encoder (CE) scores for query-item pairs in the sparse matrix takes a significant fraction (up to 90%) of overall indexing time.
> >         -  The time taken to factorize the sparse matrix is proportional to the number of query-item pairs observed in the sparse matrix. However, it remains a small fraction (less than 10%) of the overall indexing cost.
> >    - **Inference latency vs corpus size**:
> >      - Our inference strategy, AXN, yields significant improvement over existing retrieve-and-rerank pipelines even for large domains as shown in Fig 6 (Fig 7 in updated pdf) while incurring negligible overhead, as shown in Figure 5.
> >      - The only step dependent on corpus size is sampling `topk` items wrt approximate scores (line 8 in Algo 1).
> >         - In our current implementation, even a brute-force implementation of this step with 5 million items incurs negligible cost on GPUs, as shown in Fig 5.
> >         - Note that this step can also be implemented on CPUs using off-the-shelf approximate vector-based kNNs search methods if brute-force computation becomes a bottleneck.
> >
> > > Why are not all methods not represented on each sub-figure in Figure 3?
> >
> > We omitted $RnR_{TF-IDF}$ baseline for ZeShEL datasets in Fig 3a and 3b because TF-IDF baselines perform significantly
> > worse as compared to other methods (as shown in Figure 1) and adding it to the plot in Fig 3a and 3b made it difficult to clearly see differences between other better performing methods.

---

### Author Response · Authors · 2023-11-20

We would like to thank all the reviewers for their time and valuable feedback.

We have attempted to address the concerns raised by the reviewers in our response to each review. We would really appreciate an appropriate increase in the score to facilitate acceptance of the paper if the reviewer’s concerns have been adequately addressed and we would be happy to further discuss and clarify any remaining questions/concerns.

---

### Meta-Review · Area_Chair_PcRU · 2023-12-11

**Metareview:**

This paper proposes to improve the k-NN search using an adaptive method, by leveraging existing dual encoders and cross-encoders.The method appears to novel and effective.

Strength:
1. A simple yet novel method and could be used to improve many existing retrieval-rerank system.
2.  can be used to align item embeddings with the CE to improve offline indexing performance.

Weakness:
1. How to choose dual encoders and CE models might need additional efforts to make sure they can work together.
2. Although it is a simple method, it does make the retrieval-rerank method more complex as it now involves a few iterations between them. It would be nice to have some more formal analysis what the performance gap would be compared with exact topk with cross-encoders. This would be expensive, but could give some more insights.
3. The datasets used in the paper are relatively small, would be good to have some more larger-scale datasets, with hundreds of millions or billions of data points for example, where a force DE is not feasible.

**Justification For Why Not Higher Score:**

Simple yet effective method, but it could make existing approach more complex and might have limited impact on much larger-scales problems.

**Justification For Why Not Lower Score:**

An simple yet novel method that would be interesting to the relevant communities.

---

### Decision · Program_Chairs · 2024-01-16

Accept (poster)